# Spatiotemporal spike-centered averaging reveals symmetry of temporal and spatial components of the spike-LFP relationship during human focal seizures

Somin Lee[1,2,12], Sarita S. Deshpande[1,2,12], Edward M. Merricks [3], Emily Schlafly[4], Robert Goodman[5], Guy M. McKhann[6], Emad N. Eskandar[7,8], Joseph R. Madsen[7,9], Sydney S. Cash [10], Michel J. A. M. van Putten [11], Catherine A. Schevon[3] & Wim van Drongelen [1✉]

The electrographic manifestation of neural activity can reflect the relationship between the faster action potentials of individual neurons and the slower fluctuations of the local field potential (LFP). This relationship is typically examined in the temporal domain using the spike-triggered average. In this study, we add a spatial component to this relationship. Here we first derive a theoretical model of the spike-LFP relationship across a macroelectrode. This mathematical derivation showed a special symmetry in the spike-LFP relationship wherein a sinc function in the temporal domain predicts a sinc function in the spatial domain. We show that this theoretical result is observed in a real-world system by characterizing the spike-LFP relationship using microelectrode array (MEA) recordings of human focal seizures. To do this, we present a approach, termed the spatiotemporal spike-centered average (st-SCA), that allows for visualization of the spike-LFP relationship in both the temporal and spatial domains. We applied this method to 25 MEA recordings obtained from seven patients with pharmacoresistant focal epilepsy. Of the five patients with MEAs implanted in recruited territory, three exhibited spatiotemporal patterns consistent with a sinc function, and two exhibited spatiotemporal patterns resembling deep wells of excitation. These results suggest that in some cases characterization of the spike-LFP relationship in the temporal domain is sufficient to predict the underlying spatial pattern. Finally, we discuss the biological interpretation of these findings and propose that the sinc function may reflect the role of mid-range excitatory connections during seizure activity.

[1] Department of Pediatrics, University of Chicago, Chicago, IL 60637, USA. [2] Medical Scientist Training Program, University of Chicago, Chicago, IL 60637, USA. [3] Department of Neurology, Columbia University, New York, NY 10032, USA. [4] Graduate Program in Neuroscience, Boston University, Boston, MA 02215, USA. [5] Department of Neurosurgery, Lenox Hill Hospital, New York, NY 10075, USA. [6] Department of Neurological Surgery, Columbia University, New York, NY 10032, USA. [7] Department of Neurosurgery, Massachusetts General Hospital and Harvard Medical School, Boston, MA 02114, USA. [8] Nayef Al-Rodhan Laboratories for Cellular Neurosurgery and Neurosurgical Technology, Massachusetts General Hospital and Harvard Medical School, Boston, MA 02114, USA. [9] Department of Neurosurgery, Brigham and Women's Hospital and Harvard Medical School, Boston, MA 02115, USA. [10] Department of Neurology, Massachusetts General Hospital and Harvard Medical School, Boston, MA 02114, USA. [11] Clinical Neurophysiology Group, MIRA Institute for Biomedical Engineering and Technical Medicine, University of Twente, 7500AE Enschede, The Netherlands. [12] These authors contributed equally: Somin Lee, Sarita S. Deshpande. ✉email: wvandron@peds.bsd.uchicago.edu

Spatiotemporal patterns of brain electrical activity reflect neural mechanisms underpinning different brain patholo-gies. Consequently, temporal and spatial patterns observed in electrographic recordings are frequently employed to guide diagnostic and therapeutic approaches in the treatment of epi-lepsy. During surgical evaluation of patients with epilepsy, a variety of electrodes are used to record brain electrical activity across different scales. For example, large-scale (cm-range) global activity can be recorded by macroelectrodes at the scalp or cortex, and mesoscale (mm-range) and microscale (sub-mm range) activity can be recorded by intracranial arrays or bundles of microelectrodes[1–3]. Despite the heavy reliance on electro-physiology in clinical practice, the relationship between neural activity across scales and the mechanistic implications of the observed spatiotemporal patterns remain poorly characterized.

One important question in understanding cortical seizure dynamics is how the activity of individual neurons relates to local and global network activity in ictal and interictal states. The interactions of neural networks during human focal seizures across micro-, meso- and macroscopic scales have been char-acterized by other recent studies[1]. Specifically, one study showed that the spike-triggered average (STA) of the ongoing low fre-quency component of the local field potential (LFP) could be approximated by a sine cardinal (sinc) function[4]. Furthermore, filtering a train of ictal action potentials with a rectangular (brick wall) filter generated an output that correlated well with the observed seizure, consistent with the fact that the Fourier trans-form of a rectangular function is the sinc function[5]. While the ictal STA was determined in the temporal and frequency domains, the spatial component of the relationship between action potentials and low frequency LFP was not characterized.

Similarly, most previous studies that describe the relationship between single spiking activity and the surrounding LFP have focused primarily on temporal descriptions using the STA[6,7]. The few studies that have investigated the spatial component of this relationship do so by incorporating spatial information into the STA through the addition of spatial filters[8] or use a covariance-based approach[9]. None so far have directly visualized the full spatial topography of LFP associated with spiking activity.

In this study, we present a mathematical model describing an ictal spike as measured by a macroelectrode to show that in special cases, the temporal and spatial features of the spike-associated LFP can predict one another. We hypothesize that this spatiotemporal relationship is governed by the network state (ictal vs. non-ictal) and the location in the network (recruited vs. unrecruited seizure territory). To test whether this relationship can be observed in real electrographic recordings of human sei-zures, we developed an approach termed the spatiotemporal spike-centered average (st-SCA), in which the spatial topography of spike-associated LFP can be visualized by calculating a spatial average of the LFP centered around the location of individual spikes. Calculation of this topography results in a powerful tool that allows for the visualization of both the spatial and temporal components of the spike-LFP relationship. This visualization confirmed that in a subset of patients, a 1D-sinc function in the temporal domain was associated with a pattern consistent with a 2D sinc function in the spatial domain. This result suggests that in these special cases, the underlying spatial activation pattern can be inferred from temporal measurements alone. In the discussion, we propose that the temporal sinc function can be adequately described by data from sparse sampling, opening up the possi-bility that these spatial patterns can be inferred without the use of gridded microelectrode arrays. Finally, we explore the biological mechanisms and clinical implications of the observed spatio-temporal properties in the context of pharmacoresistant focal epilepsy.

## Results

**Theoretical model reveals symmetry between the temporal and spatial components of the spike-LFP relationship.** To generate a theoretical prediction of the temporal and spatial components of the spike-LFP relationship, we first introduce a mathematical model of a macroelectrode that measures the network LFP response to a single spike (Fig. 1). In this model, a single ictal action potential is generated at the center under a macroelectrode that covers a cortical surface bounded by $[-R, R]$, and the asso-ciated LFP is measured. If the spike is represented by a unit impulse, the delta function $\delta(r, \tau)$, the correlation between the spike and LFP can be described as a unit impulse response (UIR), $\mathrm{UIR}(r, \tau)$, that is governed by some function $f(r, \tau)$ of space ($r$) and time ($\tau$).

We first evaluate the UIR in the time domain, $\mathrm{UIR}(\tau)$. Because the potential of cortical generators attenuates sharply with distance, we assume that we may ignore contributions associated with the centrally located impulse in areas not directly under the macroscopic electrode. Under this assumption, the electrode's signal can be approximated by summing the contributions over only the neocortical area under the electrode:

$$\mathrm{UIR}(\tau) \approx \int_{-R}^{R} f(r, \tau)\, dr = \int_{-\infty}^{\infty} \mathrm{rect}(-R, R) f(r, \tau)\, dr \quad (1)$$

where $\mathrm{rect}(-R, R)$ represents a rectangular window bounded by $[-R, R]$.

Similarly, we can find the UIR in the spatial domain, $\mathrm{UIR}(r)$, by integration over a fixed time epoch, $[-T, T]$:

$$\mathrm{UIR}(r) \approx \int_{-T}^{T} f(r, \tau)\, d\tau = \int_{-\infty}^{\infty} \mathrm{rect}(-T, T) f(r, \tau)\, d\tau \quad (2)$$

In most cases, the underlying function $f(r, \tau)$ cannot be simply derived by measuring $\mathrm{UIR}(\tau)$ and $\mathrm{UIR}(r)$. There is a special case, however, where this derivation is possible. Note that in Eissa et al.[4], the temporal UIR of an ictal network was characterized and was shown to have the following relationship[4]:

$$\mathrm{UIR}(\tau) \propto \mathrm{sinc}(\tau) \quad (3)$$

Substituting this relationship into Eq. (1) results in the following:

$$\mathrm{sinc}(\tau) \approx \int_{-R}^{R} f(r, \tau)\, dr = \int_{-\infty}^{\infty} \mathrm{rect}(-R, R) f(r, \tau)\, dr \quad (4)$$

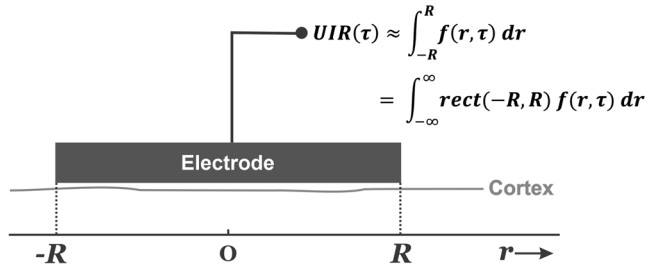

**Fig. 1 A mathematical model as represented by the recording of macroelectrode measures the underlying network's unit impulse response (UIR) to a centrally located impulse.** The electrode covers an area of one-dimensional cortex where we record the effect related with a single ictal action potential at location $r = 0$ and time $\tau = 0$, represented by the unit impulse $\delta(r, \tau)$. In this scenario, the macroelectrode measures the underlying network's temporal component of the UIR, $\mathrm{UIR}(\tau)$. This measurement can be approximated by an unknown spatiotemporal cortical function, $f(r, \tau)$ associated with the action potential, integrated over the spatial range $[-R, R]$ covered by the electrode.

Because the sinc function is the Fourier transform of a rectangular function, the relationship between time and space parallels a time-frequency Fourier-transform-pair (Supplementary Note 1 and Supplementary Fig. 1)[5]. Thus, in the special case where UIR($\tau$) is described by a sinc function, we find:

$$f(r, \tau) \propto e^{jr\tau} \tag{5}$$

The identification of $f(r, \tau)$ above now enables us to find the spatial UIR UIR($r$):

$$\text{UIR}(r) \propto \int_{-T}^{T} f(r, \tau) dt = \int_{-\infty}^{\infty} \text{rect}(-T, T) f(r, \tau) \, d\tau$$
$$= \int_{-\infty}^{\infty} \text{rect}(-T, T) e^{jr\tau} \, d\tau = \text{sinc}(r) \tag{6}$$

Thus, we find that the temporal and spatial components of the ictal UIR are symmetric and both described by sinc functions.

Note that in our model we assume that $f$ only depends on the distance $r$ from the unit impulse. Thus, while the temporal sinc function sinc($\tau$) has one dimension (time), the spatial sinc function sinc($r$) is a two-dimensional function that covers a flat surface. Supplementary Fig. 1 shows examples of 1D and 2D sinc functions.

**Characterization of the spatiotemporal spike-LFP relationship in microelectrode arrays.** Next, we asked whether the results of the above theoretical derivation could be observed in a real-world system, specifically in human seizures recorded by microelectrode arrays (MEA; Fig. 2a). To do this, we develop a calculation that we have termed the spatiotemporal spike-centered average (st-SCA). The st-SCA builds upon the more typically utilized spike-triggered average (STA) by accounting for both the timing and location of the spikes.

The spike-LFP relationship can be characterized in the temporal domain by calculating the cross-correlation $C(\tau)$ between the spiking activity and associated LFP. This cross-correlation $C(\tau)$ is mathematically equivalent to and frequently referred to as the STA[10]. To find an expression for $C(\tau)$, we first represent a multi-unit spike train with $N$ spikes occurring at times $t_i$ as a series of delta functions:

$$\sum_{i=1}^{N} \delta(t - t_i) \tag{7}$$

We then take the average LFP in a temporal window defined by a positive or negative lag $\tau$ around the spike times $t_i$. This results in the following:

$$C(\tau) = \frac{1}{N} \int_{-\infty}^{\infty} \left( \sum_{i=1}^{N} \delta(t - t_i) \right) \text{LFP}(t + \tau) \, dt = \frac{1}{N} \sum_{i=1}^{N} \text{LFP}(t + \tau) \tag{8}$$

To expand this expression to include a spatial component, we must account for both the spikes' timing ($t$) and location in the cortical plane ($x, y$). We first represent a multi-unit spike train with $N$ spikes occurring at times $t_i$ and at locations ($x_i, y_i$) as a series of delta functions:

$$\sum_{i=1}^{N} \delta(x - x_i, y - y_i, t - t_i). \tag{9}$$

We then take the average LFP in a temporal window defined by a positive or negative lag $\tau$ around spike times $t_i$ and a spatial window defined by the plane ($\xi, \psi$) around locations ($x_i, y_i$). This produces the expression for the normalized spatiotemporal cross-correlation $C(\xi, \psi, \tau)$ between the LFP and action potential:

$$C(\xi, \psi, \tau) = \frac{1}{N} \left( \sum_{i=1}^{N} \delta(x - x_i, y - y_i, t - t_i) \right)$$
$$\text{LFP}(x + \xi, y + \psi, t + \tau) \, dx \, dy \, dt \tag{10}$$

To evaluate this expression, we interchange the integration and summation operations and integrate over the spatiotemporal domain. The resulting expression is what we have termed as the spatiotemporal spike-centered average (st-SCA):

$$C(\xi, \psi, \tau) = \frac{1}{N} \sum_{i=1}^{N} \text{LFP}(x_i + \xi, y_i + \psi, t_i + \tau) = \text{st} - \text{SCA}(\xi, \psi, \tau) \tag{11}$$

Note that if we set the range of ($\xi, \psi$) equal to the area covered by a fixed spatial range, we obtain the well-known temporal STA as in Eq. (1) (Fig. 2b). In contrast, if we set $\tau$ to a fixed temporal range, we obtain purely the spatial component of the st-SCA for that epoch, conform Eq. (2). The mathematic relationship between the st-SCA and the UIR is discussed in detail in Supplementary Note 2. In the following, we describe the computational steps to determine the st-SCA in MEA recordings.

**Application of st-SCA to clinical microelectrode array recordings.** To apply the st-SCA to microelectrode array (MEA) recordings, we must account for the irregular timing and location of spiking activity across the array. A simplified analogy of this approach is to visualize spiking activity as stones being tossed into water. Consider throwing a single stone into water and analyzing the consequent effects by observing the resulting water ripples. We can simulate multiple sources by dropping identical stones from the same height but at different times and locations across the horizontal plane of the water surface, resulting in a complex landscape. To determine the contribution of a single stone to this landscape, we can take a field of view centered around individual stones. According to Eq. (11), averaging across all stones gives the stone's characteristic spatiotemporal perturbation.

To apply this to the analysis of MEA recordings, spikes are detected for each channel in the MEA and the low frequency LFP associated with each spike is determined (Fig. 2c). This LFP is then spatially translated such that the associated spike position ($x, y$) is at the origin of a new axes ($\xi, \psi$) (Fig. 2c). This spike detection and LFP translation process is then applied to all channels. Averaging the results across all channels results in a field of view of the spike-associated LFP that is (1) centered around individual spikes and (2) approximately four times larger than the area of the MEA (Fig. 2c). This field is then calculated for time points $\tau$ to result in the st-SCA.

**Visualization of spatial and temporal components of the spike-LFP relationship.** We applied the st-SCA method to analyze microelectrode recordings of 19 focal seizures across seven patients undergoing epilepsy surgery evaluation (Supplementary Table 1). These recordings were obtained from 96-channel, $4 \times 4$ mm Utah microelectrode arrays (MEA)[3,11] (Methods). Although the MEA was implanted in the seizure onset zone as determined during clinical assessment for all patients, five patients were determined to have arrays implanted in recruited seizure territory (Patients 1–5), and two patients had arrays implanted in unrecruited seizure territory (Patients 6–7) (Fig. 2a). As previously described[3,11], recruited seizure territory is an area of tissue that is invaded by the ictal wavefront throughout the course of a seizure. The ictal wavefront is defined by high rates of firing that is highly correlated with overlying low frequency rhythms. Unrecruited territory sees no invasion of the ictal wavefront but still shows rhythmic EEG activity due to local

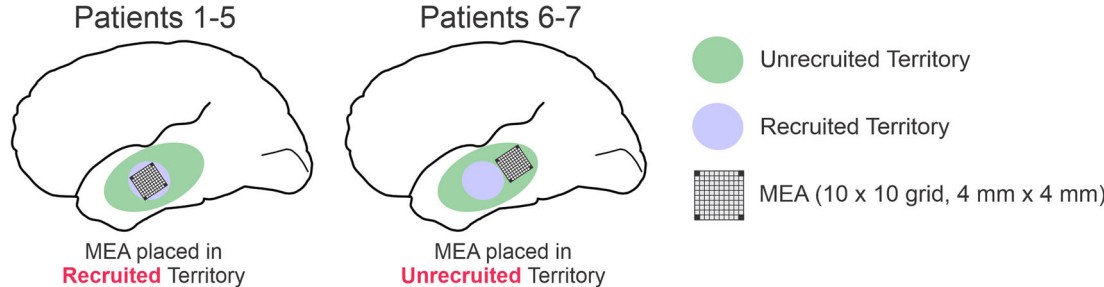

## a) Example Placement of Microelectrode Array (MEA)

Schematics not drawn to scale

Patients 1-5 — MEA placed in **Recruited** Territory

Patients 6-7 — MEA placed in **Unrecruited** Territory

Unrecruited Territory / Recruited Territory / MEA (10 x 10 grid, 4 mm x 4 mm)

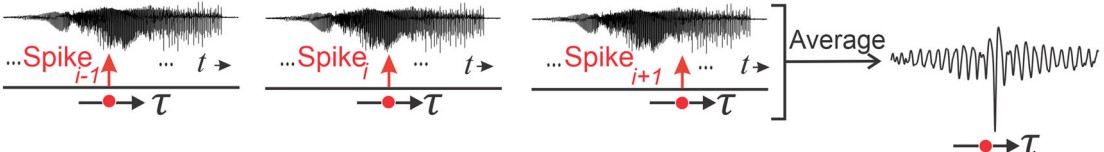

## b) Seizure Activity and the Temporal Spike-Triggered Average

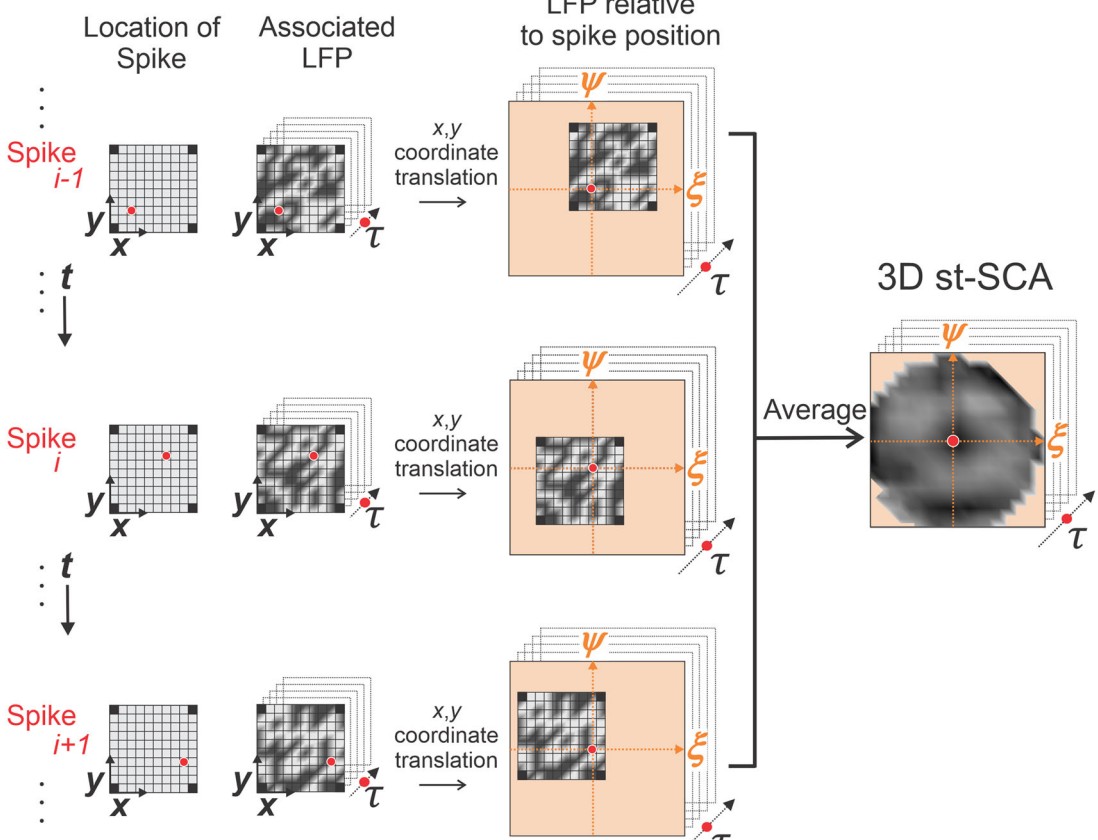

## c) Spatiotemporal Spike-Centered Average (st-SCA)

Location of Spike — Associated LFP — LFP relative to spike position

Spike $i-1$ / Spike $i$ / Spike $i+1$

$x,y$ coordinate translation

Average — 3D st-SCA

synaptic activity[3,12]. Both ictal and interictal recordings were evaluated, where interictal was defined as being at least 2 h away from any known ictal activity. Filtering was used to extract local multi-unit neural firing activity and the associated low frequency component of the LFP of the surrounding network (Methods).

Representative ictal and interictal signals and STAs calculated across ictal and interictal states for two recruited territory recordings and one unrecruited territory recording are depicted in Fig. 3. The black lines in Fig. 3c–h represent the STAs, the red lines represent the associated noise estimates, and the vertical dotted lines indicate $t = 0$, i.e., the timing of the spike trigger. For both recruited and unrecruited territory recordings, the amplitudes of the ictal STAs (Fig. 3c, e, g) were larger than the corresponding the interictal STAs (Fig. 3d, f, h). The amplitude for the unrecruited ictal STA (Fig. 3g), however, was much smaller than the recruited ictal STAs (Fig. 3c, e).

Patients with the MEA located in recruited seizure territory showed STAs with different morphologies (Supplementary

**Fig. 2 The method to compute the mesoscale spike-triggered average and spatiotemporal spike-centered average (st-SCA) between spiking activity and the low frequency component of the local field potential (LFP) during a human focal seizure involves centering and averaging each spike's associated LFP in both time and space. a** Diagram of the microelectrode array (MEA) placement, a 10 × 10 grid of electrodes of 4 mm × 4 mm in size, was implanted in either recruited territory (blue shading; Patients 1–5) or unrecruited territory (green shading; Patients 6–7). Recruited territory involves a seizure passing through and invading the local cortical tissue, and unrecruited territory is tissue outside the recruited territory but may still be characterized by strong, local synaptic activity[3,12]. **b** During seizure activity, the LFPs within the area of the electrode array (the summed LFP of the microelectrode array is depicted in the signal traces) are associated with a multi-unit action potential train. The LFP's relationship to the spike is considered over time $\tau$ relative to the spike events. **c** For each spike across the MEA, its associated spatiotemporal LFP is determined. The red circle in the middle column indicates the spike position on the MEA. Next, the $(x, y)$ axes of the LFP are translated into the $(\xi, \psi)$ axes, such that the spike position is at the origin. Finally, the results in the right column are averaged to create a matrix that contains the st-SCA. Note that the corners of the average are undefined because the MEA does not have electrodes in the corner positions.

Fig. 2a–e), but all had a dominant negative peak around $t = 0$. Consistent with previous findings, we found that the STA for Patients 1–3 resembled a sinc function with a peak embedded in a weak oscillatory component (Figs. 3c and 4c and Supplementary Fig. 2a–c)[4]. In contrast, the STA for Patients 4 and 5 did not resemble a sinc function as Patient 4 showed a dominant peak embedded in a strong oscillation (Supplementary Fig. 2d) while Patient 5 showed no oscillatory component (Figs. 3e and 4d and Supplementary Fig. 2e). The STAs calculated from MEAs implanted in unrecruited territory (Patients 6 and 7) were weak with a smaller amplitude deflection around to $t = 0$ (Fig. 3g and Supplementary Fig. 2f, g).

We then evaluated the relationship between spiking activity and the LFP in the spatiotemporal domain by computing the st-SCA over the entire MEA $(\xi, \psi)$ and times $\tau = \pm 1$ms (Fig. 4e, f). This 2 ms interval averaged across to yield a 2D spatial topography. In the ictal phase for Patients 1–3, we observed a centrally located trough surrounded by a pair of rings with apparent radial symmetry, a shape that is consistent with the center of a 2D sinc function (Fig. 4e and Supplementary Fig. 3a). The distance between the center and the region indicated by the inner circle was ~1.5 mm, and the distance between center and the region indicated by the outer circle was ~2.5 mm (Fig. 4e). In contrast, the ictal phase for Patients 4 and 5 showed a deep well of stronger negative activity (Fig. 4f and Supplementary Fig. 2d, e). The st-SCAs during the interictal phase as well as the results obtained in unrecruited territories showed different patterns with relatively smaller amplitude signals (Supplementary Fig. 3b, d, e, f). The temporal and spatial results across seizures within each patient were consistent. Representative st-SCAs for each patient are depicted in Supplementary Fig. 2. Representative noise estimates are shown in Supplementary Fig. 4.

In sum, three of five patients with recordings from recruited territories (Patients 1–3) showed STAs with sinc function morphology and st-SCAs with a donut-ring pattern that was consistent with the center of a 2D sinc function (Fig. 4c, e and Supplementary Fig. 2a–c). In Patients 4 and 5, the STAs did not have a sinc morphology, and the st-SCAs showed deep and diffuse wells of negative activity (Fig. 4d, f and Supplementary Fig. 2d, e).

Note that these observations were not attributable to widespread correlations among MEA electrodes. To demonstrate that the observed st-SCA patterns are representative of the spike-LFP relationship and not the global correlations among network LFPs, we showed that STAs in unrecruited territory show a large and significant oscillatory component only when triggered by spikes from recruited territories (Supplementary Fig. 5b), and not when triggered by spikes from unrecruited territories (Supplementary Fig. 5c). This result is a replication of previous studies[1]. Furthermore, randomizing the spike times detected across the MEA resulted in complete destruction of the observed st-SCA patterns, emphasizing the importance of spike timing as the driver for these spatiotemporal patterns (Supplementary Fig. 6).

Finally, calculation of the st-SCA after applying a spatial filter to decorrelate LFP signals across MEA channels did not qualitatively alter the st-SCA patterns (Methods, Supplementary Fig. 7).

**Quantification of the peak-to-peak distance of spatial patterns.** Next, we aimed to more quantitatively describe the donut-shaped activity observed in Fig. 4e. Taking advantage of the qualitatively observed radial symmetry observed in the st-SCA, we converted the Cartesian coordinates $(\xi, \psi)$ into polar coordinates $(r, \theta)$ and focused on the spatial relationship with respect to $r$ (Fig. 5a). This enabled us to depict the st-SCA in two dimensions, $(r, \tau)$ (Fig. 5b), similar to function $f(r, \tau)$ in Fig. 1. A detail of that relationship is depicted in Fig. 5c, and the summed values across this two-dimensional detail are plotted along its margins. These summed values are the two components as a function of space and time ($r$ and $\tau$). Note that the graph in the bottom margin of Fig. 5c represents the central trough ($\tau = \pm 35$ms) of the function shown in Fig. 4c. As anticipated by the outcome in Eq. (6) we observed a spatial component (Fig. 5c) that shows a central trough with smaller amplitude side lobes—a pattern consistent with the shape of a sinc function. Note that the resolution and range of the spatial component ($r = \pm 3.6$mm) is limited by the size of the MEA (Methods, Fig. 2c). Consistent with the donut-shaped rings observed in Fig. 4e, the peaks of the function shown in the left margin of Fig. 5c were separated by ~2.5 mm (blue arrows, Fig. 5c).

## Discussion

A key result of this study is that there exists a mathematical symmetry between the temporal and spatial domains of the spike-LFP relationship in the special case where both domains resemble a sinc function. By characterizing the spatiotemporal components of the spike-LFP relationship in microelectrode array (MEA) recordings of human focal seizures, we showed that this mathematical symmetry is not confined to the theoretical realm. Of the five patients with recruited territory recordings, three showed temporal and spatial patterns consistent with this symmetric relationship. The existence of this symmetry in clinical recordings offers an interesting implication: in some cases, the underlying spatial pattern of the spike-LFP relationship may be inferred by characterization of the temporal pattern alone. Specifically, a sinc pattern observed in the temporal domain predicts a 2D sinc pattern in the spatial domain (Eqs. (4)–(6), Figs. 1 and 6a).

This predictive power is important in the context of clinical microelectrode recordings because it suggests that it may be possible to characterize spatial patterns without the use of gridded MEAs. While MEAs are advantageous for monitoring and studying seizure activity with high temporal and spatial resolution, their current clinical utility is limited as they cannot be easily used to sample from multiple cortical areas. Interestingly, we found that the sinc function can be characterized in the temporal domain by using spiking and LFP information from a random subset of only eight electrodes (Fig. 6b). Although the spatial

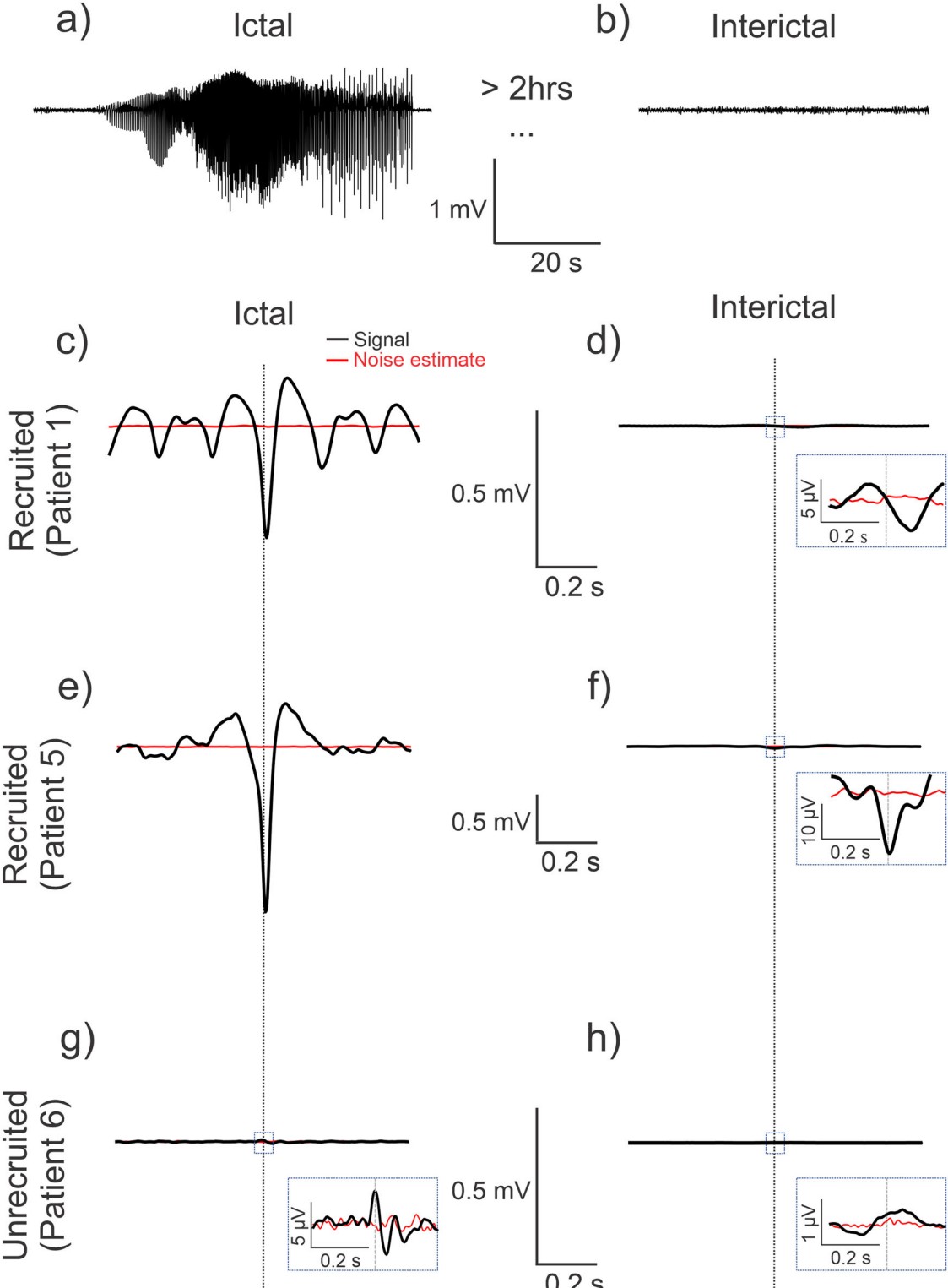

**Fig. 3 Spike-triggered averages (STAs) in recruited and unrecruited cortical territories during ictal and interictal phases show different patterns.** The black traces are the signals, and the red traces represent the associated noise estimates. Vertical stippled lines represent the zero of the time-axis. **a**, **b** Example signal trace of average ictal and interictal LFP activity across MEA channels. **c–f** The STA (black trace) in the recruited territories show an evolution toward a characteristic negative peak, with or without surrounding oscillations, during the ictal phase. The ictal phase amplitudes are also much higher than those of the interictal phase. Noise estimates are shown by the red traces. **g**, **h** The results in the unrecruited territory show comparatively low amplitudes as compared to the recruited STAs.

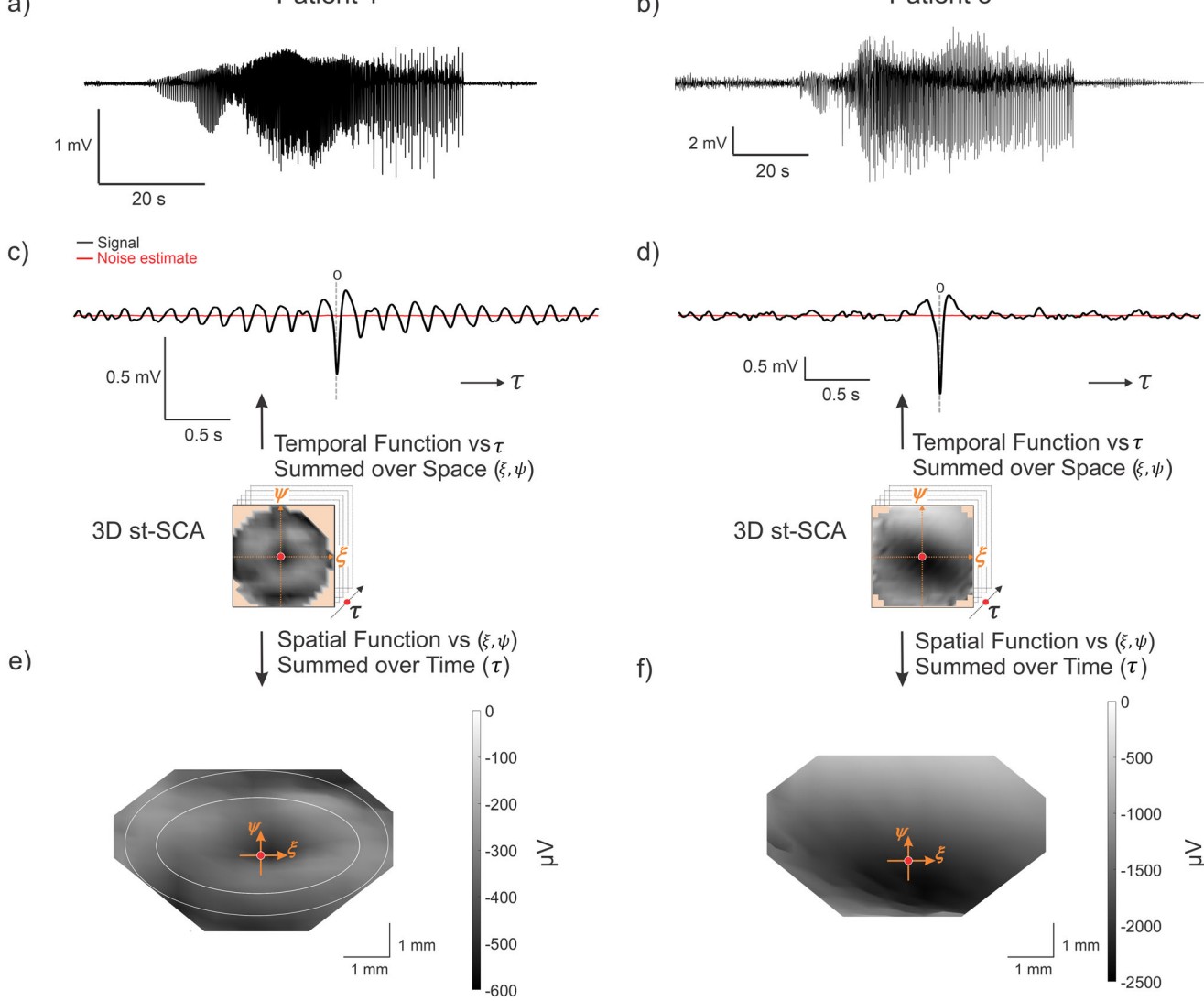

**Fig. 4 Properties of the ictal spatiotemporal spike-centered average (st-SCA) function during a focal seizure show different patterns for two representative patients with the MEA positioned in recruited territory. a, b** Representative traces of the average LFP activity across the microelectrode array for Patient 1 and Patient 5. **c, d** The temporal average (black trace) is calculated by averaging the st-SCA over all spatial contributions (±3.6 mm). Noise estimates are shown in red. **e, f** A 3D view (azimuth = 0°, elevation = 70°) of the 3D st-SCA summed over time $\tau = \pm 35$ ms. The center ($\xi, \psi = 0, 0$) is indicated by the red dot. The two concentric circles in (**e**) are drawn to indicate that the center is surrounded by two rings. Note the apparent radial symmetry of the st-SCA pattern in Patient 1. The $\xi$-axis and $\psi$-axis represent the spatial dimensions of the MEA, and the third dimension ($z$-axis) in this topological view represents microvolt (μV) units. The grayscale corresponds to the $z$-axis and is in μV units.

pattern is impossible to discern with just eight electrodes, the underlying spatial pattern may be inferred to be a 2D sinc function since the associated temporal pattern is a sinc function. This suggests that the st-SCA may be characterized by using neocortical microelectrodes that allow for recording from multiple areas by reducing the number of channels per probe. The development of such electrodes is technologically feasible as similar probes are already used clinically for the monitoring of deep brain structures such as Behnke-Fried depth electrodes[13].

A natural and necessary question to ask at this junction is whether the 2D sinc function in the spatial domain has any biological meaning. While the exact mechanisms underlying temporal and spatial sinc patterns are beyond the scope of this study, we propose here that the concentric donut-ring pattern in the spatial component of the spike-LFP relationship may reflect the engagement of mid-range horizontal connections during seizure initiation and propagation.

Under physiological conditions, synaptic activity is a major contributor to the extracellular potential field[14]. Other contributors may include intrinsic membrane currents, gap junctions, neuron-glia interactions, and ephaptic effects[15,16]. While the relative contributions of these different mechanisms during pathological states such as seizures have not been fully elucidated, a non-zero cross-correlation between action potentials and LFPs is expected because synaptic currents are a major component in the compound activities observed in ictal states.

In our discussion of the biological implications of the observed st-SCAs, we adopt the interpretation for our specific electrode configuration as previously described[1] by assigning a net excitation to negative deflections and net inhibition to positive deflections. This interpretation is also in line with previous studies of the ictal core and propagation[3,17]. Accordingly, our st-SCA analyses (Figs. 3 and 4) show that in the recruited ictal territory, the spike-LFP correlation at small lags is dominated by

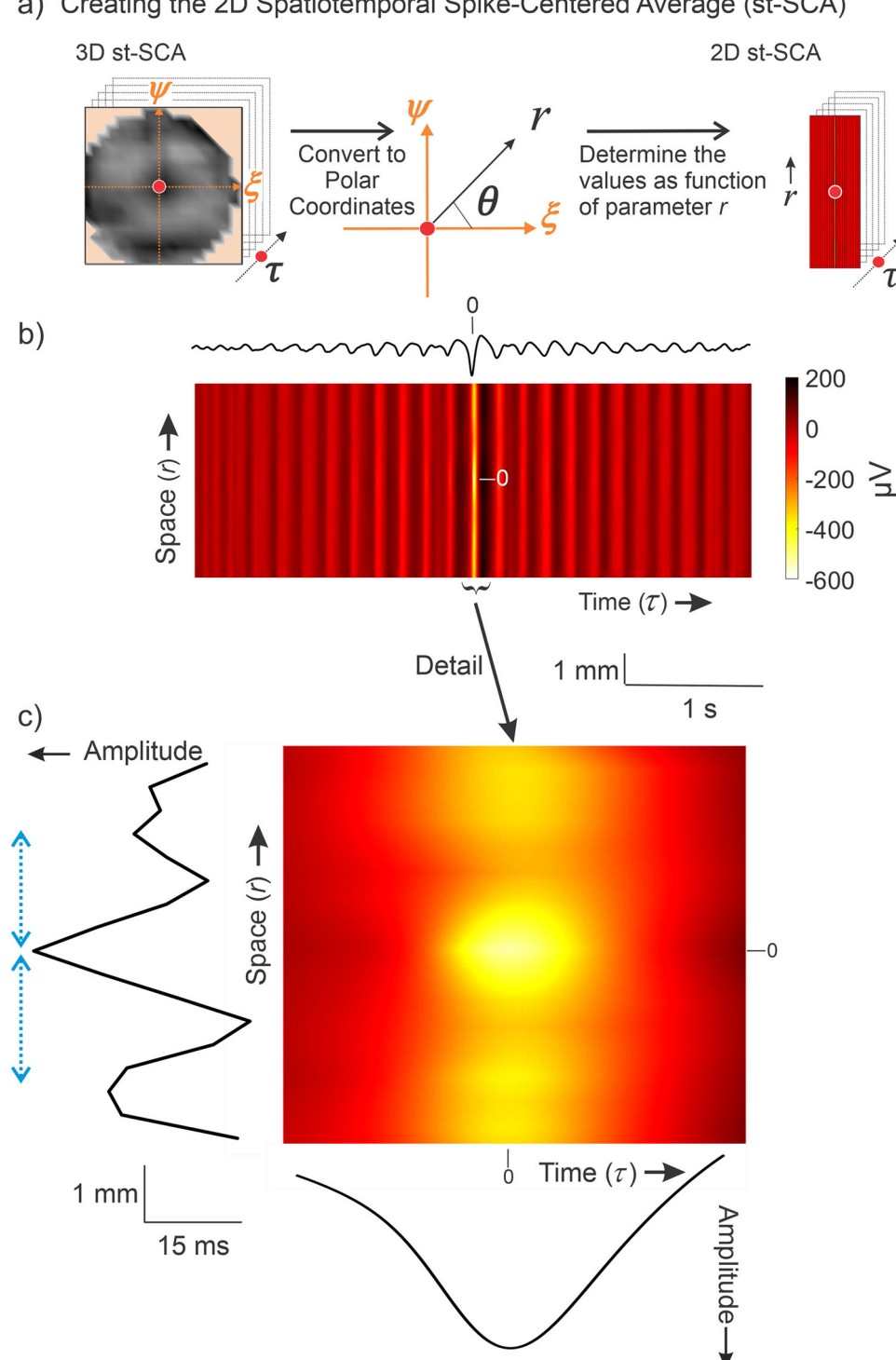

**Fig. 5 The method to compute the 2D spatiotemporal spike-centered average, st-SCA($r$, $\tau$) allows for a detailed visualization of spatial and temporal components for Patient 1. a** The Cartesian coordinates ($\xi$, $\psi$) from the 3D st-SCA are converted into polar coordinates ($r$, $\theta$), resulting in a 2D st-SCA. **b** A color representation of st-SCA($r$, $\tau$). The temporal component of the st-SCA($\tau$) (black trace) is obtained by the sum of st-SCA($r$, $\tau$) over $r$ (same as the signal in Fig. 4e). Amplitude and color scales are in µV. **c** Detail of the central part of (**b**). The left margin shows the resulting wave from summation over time, generating the spatial component of st-SCA. The blue arrows on the left indicate the distance (~2.5 mm) between the peaks seen in this function. The bottom margin depicts the resulting wave from summation over space, generating the temporal component of the st-SCA.

net excitation during seizures in all patients. The activity level in the excitatory center, representing the activity at the ictal wave, is excessively high, possibly due to saturation of the local inhibitory population[17]. In Patients 1–3 we also observe a ring of reduced excitation at a distance ~1.5 mm around the excitatory center

(Fig. 4e and Supplementary Fig. 2a–c). In turn, the ring of reduced excitation is surrounded by a second ring at an additional distance of ~1 mm where excitation increases again. For these patients, this donut-shaped st-SCA is specific to the recruited seizure territory in the ictal phase (Supplementary Figs. 2a–c

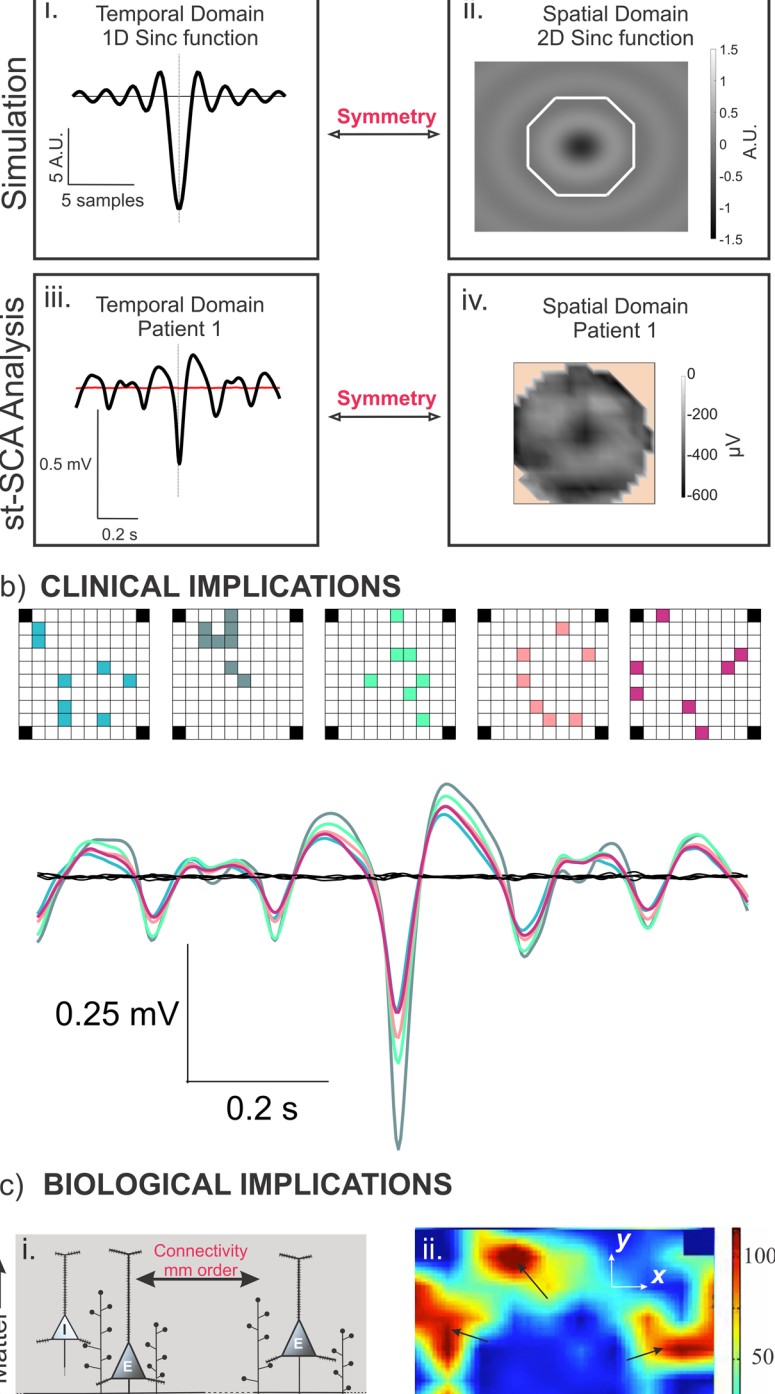

and 3a). This observation suggests that the ictal wave in the recruited territory, represented by the excitatory center $(\xi, \psi = 0, 0)$, creates an escape of hyperexcitation via a jump that engages mid-range connectivity in the millimeter range. Decorrelation of the LFP prior to the st-SCA calculations yielded similar spatiotemporal patterns (Supplementary Fig. 7), further

corroborating the importance of local millimeter range excitatory connections in focal seizures[8].

A question that remains is whether there is any biological evidence that supports this type of connectivity. Histological studies have shown that there are indeed excitatory mid-range connections at the millimeter scale mediated by axon collaterals

**Fig. 6 Summary of key findings and biological/clinical implications of results. a** There exists a special mathematical symmetry between the temporal and spatial domains if they resemble sinc functions. **a**i and **a**ii show simulated results, and **a**iii and **a**iv show results from the clinical st-SCA analysis for Patient 1. Note that the white octagon in the simulated 2D sinc function (**a**ii) represents the limited size of the MEA (as we are unable to see beyond the octagon due to the 4 mm × 4 mm shape of the MEA). However, a 2D sinc function extends beyond this octagon as shown. **b** Spike-triggered average (STA) calculated from using spike timing and LFP activity from a random subset of eight electrodes for Patient 1. The sinc function may be characterized in the temporal domain using signals from only eight channels across the MEA. The different colors represent different random subset of eight electrodes, and their corresponding STA. **c** The propagation of the ictal wavefront supports the involvement of mid-range excitatory neocortical connectivity by axon collaterals. **c**i Diagram of gray matter excitatory connections of a neocortical pyramidal cell showing the short-range connections (order of 100 s of μm) and mid-range connections (order of mm) via the pyramidal cell axon collaterals (based on Fig. 5 in Nieuwenhuys[18]). Excitatory neurons are labeled "E," and the inhibitory neuron is labeled "I" in this schematic. **c**ii Snapshot of Supplementary Movie 1 depicting the propagation of ictal multi-unit action potentials across part of a Utah array. The black arrows show multiple contiguously active areas that are separated by a mid-range mm-sized distance, supporting that the excitatory axon collateral connections are invoked for propagation of the ictal activity. Color scale represents the number of spikes per second.

within the gray matter in the neocortex in addition to short-range excitatory and inhibitory connections at a scale of hundreds of μm (Fig. 6c)[18–21]. Additionally, previous studies of ictal wave dynamics provide direct evidence that mm-range connections are invoked during seizure activity[3]. An example of this jump in action potential activity is depicted in the spatial plot in Fig. 6c (a snapshot of Supplementary Movie 1), in which there are multiple areas of simultaneously increased neural activity across the MEA, separated by mm-range gaps. This is consistent with the distance between the excitatory center and outer ring we observe in the donut-shaped spatial cross-correlation depicted in Fig. 4e. This pathological escape of uncontrolled excitation across cortex could be considered a candidate mechanism in seizure recruitment and propagation (Supplementary Note 3).

Not all patients with implants in recruited territory showed spatiotemporal patterns resembling a sinc function, and the clinical etiologies for these patients may offer some clues about why this is the case. The diffuse depressions observed in the spatial domains for both Patients 4 and 5 (Fig. 4f and Supplementary Figs. 2d, e and 3c) seem consistent with a local flood of excitation. Indeed, the seizures in both of these patients were characterized as secondarily generalized (Supplementary Table 1). This suggests that in generalized seizures, the mid-range excitatory connectivity structure (as represented by the sinc function) may play a diminished role in comparison to other mechanisms of ictal propagation, such as local excitation or engagement of white matter tracts (Fig. 6c). Furthermore, a unique case is Patient 3, who was diagnosed with cortical dysplasia (Supplementary Table 1). The STA is sinc-like, and the st-SCA partially resembles a sinc function (Supplementary Fig. 2c). Cortical dysplasias have been shown to be associated with functional connectivity defects[22–24], which may explain the partial donut ring of activity in the st-STA (Supplementary Fig. 2c). If indeed st-STA patterns reflect underlying pathologies, clinicians could potentially use these spatiotemporal characterizations to target specific mechanisms underlying a patient's seizures and choose appropriate therapeutic strategies. For example, removal of horizontal interactions on a mm-scale has been the rationale for performing subpial transections in patients with intractable epilepsy[25]. In these cases, characterization of the st-SCA may inform the appropriateness of such interventions in personalized patient treatment plans. Furthermore, because this method includes the spatial domain, the st-SCA method can be used for a broad scope of applications, such as MEA cultures[26,27], Utah arrays implanted in monkeys completing tasks[28–30], MEAs implanted in humans for sleep[31], and for brain-computer interfaces[32,33].

## Methods

**Patients**. Seven patients with pharmacoresistant focal epilepsy underwent chronic intracranial EEG studies to help identify the epileptogenic zone for subsequent removal. Patients 1, 4, 6, and 7 were recruited at Columbia University Medical Center, and Patients 2, 3, and 5 were recruited from Massachusetts General

Hospital/Brigham and Women's Hospitals (Supplementary Table 1). The entirety of the ictal segments and 2-min interictal segments were used in these analyses (Supplementary Table 1). Procedures were approved by the Internal Review Board committees at Columbia University Medical Center, The University of Chicago Comer Children's Hospital, and Massachusetts General Hospital/Brigham and Women's Hospitals. The patients' surgeries and treatment plans were not directed by or altered as a result of these studies. All ethical regulations were followed, and all patients provided informed consent regarding the use of their data for research purposes.

**Signal acquisition and pre-processing**. A 96-channel, 4 × 4 mm MEA (Utah array; Blackrock Microsystems) was implanted into neocortical gyri along with subdural electrodes (ECoG). The 96 microelectrodes were 1 mm in length and arranged in a regular 10 × 10 grid pattern with empty corners, defaceterized prior to implantation. Expanded and complete details of study enrollment, clinical evaluation of the SOZ, surgical procedures, and recording parameters have been previously published[3,34]. Signals from the MEA were acquired continuously at a sample rate of 30 kHz per channel (0.3–7500 Hz bandpass, 16-bit precision, range ±8 mV). The reference was epidural. Up to three seizures from each patient were selected for detailed analysis to avoid biasing the dataset from the patients from whom many seizures were recorded. Seizure recordings were categorized as recruited or unrecruited territory following the results of a previously published study (Fig. 2a)[3]. Briefly, seizure recordings in which the low frequency LFP correlates with underlying spike activity are considered as recruited territory recordings, and seizure recordings in which the low frequency LFP shows no correlation with underlying spiking activity are considered unrecruited territory recordings[3]. Channels and time periods with excessive artifact or low signal-to-noise ratio were excluded. Recordings were obtained during the presurgical evaluation of the patients.

Unit activity was identified using filtered 0.3–3 kHz signals with spikes defined as deflections ≥4 standard deviations below the mean. The low frequency component of the local field potential (LFP) activity across the array was created by averaging the artifact-free LFP activity from all microelectrode signals filtered 2–50 Hz. The averaged LFP procedure has been shown to generate signals that are representative of and comparable to nearby electrocorticography signals[1,4].

**Spatiotemporal spike-centered average (st-SCA) calculations and signal analysis**. All signal processing and statistical analyses were performed in MATLAB (MATLAB, Natick, MA, USA). The spatiotemporal spike-centered average (st-SCA) was determined using the following steps (Fig. 1). Each broadband signal of the 10 × 10 MEA was bandpass filtered for the low frequency component (2–50 Hz) of the local field potential (LFP) and for spike detection (0.3–3 kHz)[1,4]. Spikes were detected in the multi-unit activity as negative deflections that exceeded four standard deviations of the filtered signal. A complete list of spike detection results can be found in Supplementary Table 2. For each spike the 10 × 10 frames of the LFP data were collected for ±n sample times representing ±5 s around the spike time, and the timescale of the frames was set such that the spike occurred at time zero, $\tau = 0$. All LFP frames associated with a single spike were translated such that the spike location was at the origin of the new spatial coordinate system $\xi, \psi = 0, 0$. Note that this spatial translation is necessarily spike specific because spikes do occur at different locations. Next, the translated 10 × 10 × (2n + 1) frames were put into a three-dimensional 19 × 19 × (2n + 1) configuration with the spatiotemporal origin ($\xi, \psi, \tau = 0, 0, 0$) is at position 10, 10, n + 1. This step was done to keep the LFP frames compatible across spikes. For each spike, these frames were summed into a three-dimensional 19 × 19 × (2n + 1) matrix. For each position in the 19 × 19 × (2n + 1) matrix, the total number of contributions N was counted. Finally, to obtain the spatiotemporal cross-correlation, the sum obtained in step 6 was divided by the N obtained in step 7 for each position. This resulted in the discrete spatiotemporal estimate of $C(\xi, \psi, \tau)$, as shown in Eq. (3). This method to determine spatiotemporal patterns is based on a spike trigger that is not constrained spatially because an ictal action potential can occur across the spatial dimension of the MEA.

Evidence of radial symmetry of the st-SCA (Fig. 4e) allowed conversion from Cartesian coordinates $(\xi, \psi)$ coordinates to polar coordinates $(r, \theta)$. By ignoring the minor deviations from radial symmetry, we focused on the spatial component of the st-SCA with respect to $r$ (Fig. 5a), which enabled us to depict the spatiotemporal properties in two dimensions (Fig. 5). Furthermore, if we compute the sum across space, we obtain purely the temporal component of the st-SCA, which is equivalent to the STA. Similarly, summation over time $\tau$ generates the spatial component of the st-SCA. With these results, we can assess to what extent our model of the ictal network, a linear time-invariant (LTI) system with unit impulse response $C(\tau) \propto \text{sinc}(r, \tau)$, fits the data.

**Spatial filtering**. For calculations involving the spatial filtering of LFP signals, we applied the spatial whitening process as described in Hyvärinen et al.[35] and Telenczuk et al.[8]. As previously published, a signal is spatially filtered by matrix multiplication with a whitening matrix $W$, where $W$ is the inverse square root of the signal's covariance matrix, $C$:

$$W = C_{\text{signal}}^{-1/2} = ED^{-1/2}E^T \tag{12}$$

where $E$ is a matrix of eigenvectors of $C_{\text{signal}}$, and $D$ is a diagonal matrix with inverse square roots of eigenvalues $\lambda_i$ on its diagonal, such that $D_{ii} = \frac{1}{\lambda_i}$ and $D_{ij} = 0$[8]. In this study, the signals being transformed were the MEA channel signals bandpass filtered at 2–50 Hz.

**Statistics and reproducibility**. The number of analysis segments (both ictal and interictal) per patient, epoch length, and number of spikes are shown in Supplementary Table 2. Replicates are defined by the number of seizures per patient.

The signal-to-noise ratio (SNR) was computed for each st-SCA by estimating the residual noise using the plus-minus averaging approach. We implemented this by employing the above eight steps while keeping two three-dimensional $19 \times 19$ matrices: one summed the even contributions for each location and the other summed the odd ones. To obtain the averages for the odd and even components, each position in the matrix was then divided by its number of contributions. The sum of the even and odd averages is the same result obtained in step 8 above. In contrast, the difference between the even and odd averages cancels the consistent component (i.e., the signal) while preserving the random noise estimate. The SNR was estimated by computing the root mean square (rms) of the signals and the rms of their noise estimates, leading to a signal-to-noise ratio, $\text{SNR} = 20 \log(\frac{\text{rms}_{\text{signal}}}{\text{rms}_{\text{noise}}})$ dB. Average ratios for the st-SCAs across space and time all were >30 dB.

An example of the total number of contributions per each position in the $19 \times 19$ grid is shown in Supplementary Table 3. The average values in each pixel of the $19 \times 19$ image is based on variable numbers of trials, with less trials toward the corners of the picture (e.g., Supplementary Table 3). Due to these unequal number of contributions, the signal-to-noise ratio also varies across the image (Supplementary Table 4), but with the exception of very few pixels (four in Supplementary Table 4), all locations satisfy the so-called Rose criterion, namely that in order to distinguish image features reliably, the signal's amplitude must be 4–5 times (12–14 dB) the amplitude of the associated noise[36,37]. Finally, in our spatial images, we excluded the corners of the image from our analyses because there are no observations these parts of the image (NaN in Supplementary Table 4).

**Reporting summary**. Further information on research design is available in the Nature Portfolio Reporting Summary linked to this article.

## Data availability
The data used in this study are not publicly available due to IRB protocols at the University of Chicago, Columbia University, and Harvard University and all data sharing will have to follow protocols compliant with registered IRB protocols. Questions about data availability can be addressed at the corresponding author (W.v.D.).

## Code availability
All scripts and programs used to generate the results in these data are publicly available in a Github repository (https://github.com/sominlee14/stSCA_scripts) as well as Zenodo[38].

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

## Acknowledgements

We thank Drs. Mark Kramer, Stephan A. van Gils, Hil Meijer, Douglas R. Nordli Jr., and Jack D. Cowan for valuable discussion and suggestions. C.A.S. and W.v.D. were supported by NIH Grants R01 NS095368 and R01 NS084142. C.A.S., S.S.C. and E.S. were partially supported by NIH Grant R01 R01NS110669. S.L. and S.S.D. were supported by University of Chicago MSTP Training Grant T32GM007281.

## Author contributions
S.L., S.S.D., M.J.A.M.v.P., C.A.S., E.M.M. and W.v.D. designed the research. S.L., S.S.D., E.M.M., E.S., E.N.E., J.R.M., S.S.C., R.G., G.M.M., M.J.A.M.v.P. and W.v.D. performed the research. S.L., S.S.D. and W.v.D. performed the data analysis. S.L., S.S.D., E.M.E., J.R.M., S.S.C., M.J.A.M.v.P., C.A.S. and W.v.D. wrote and edited the manuscript.

## Competing interests
The authors declare no competing interests.
