## [Peer Review File · Communications Biology]

Reviewers' comments:

Reviewer #1 (Remarks to the Author):

This manuscript introduces a method for linking multi-unit spiking activity, and local field potentials recorded in high density invasive cortical recordings through spatiotemporal averaging. This work expands on a series of prior publication that focus on the temporal correlation (through spike triggered averages) and the same dataset of seizure recordings performed using multi-electrode arrays. The paper is focussed on introducing the method itself, and does so with clarity and nicely chosen illustrating examples and metaphors. Given the heterogeneous nature of the small patient cohort, it is difficult to assess how much generality specific observations (like the ring shape) hold beyond the studies sample. However, the method is certainly a relevant expansion around the more commonly used spike-triggered averaging, and the paper - with mostly only minor suggestions from my end - would make a great addition to the literature. I also appreciate the authors for making their code so easily available on github.

Major comments

1) How does the suggested method differ from other approaches that translate spatiotemporal patterns into 3D maps (e.g. statistic parametric mapping performed on ERP data - e.g. Stefanics et al. 2018 <https://www.jneurosci.org/content/38/16/4020/tab-figures-data> - Fig 3) - I'm assuming the answer here lies in the ability to generate maps for several concurrently occurring spiking patterns? Could the same be achieved by generating statistic parametric maps on transformed coordinates for each spike location? Would these two approaches be equivalent? How can the authors' approach be used to make statistical inference regarding the observations they have described in the aper?

Minor comments

1) The reference in line 72 as given refers to a whole book - it would be helpful to refer the reader to the specific section/chapter that supports the statement made

2) it would be useful to define 'mesoscale' in spatial terms as used here for the reader - I see these terms are later mapped in Figure 6, but this would be helpful earlier in the manuscript text

3) Figure 3 - it is unclear to me what the z-axis is on this plot? xy are in mm so presumably spatial - does z represent uV of the signal that is also colour coded?

4) providing an example plot of the sinc function would be useful

5) In the discussion, the authors mention that the secondary generalisation may contribute to differences in observation - did the authors use the whole seizure, or segments of the seizure for their analysis? If the latter, it would be useful how these relate to time of generalisation ?

6) Figure 6A - it is not quite clear what the different cells / dendritic trees are / what the horizontal line represents? More labels could be useful

7) Could a representative trace of the LFP signal be included for the two patients illustrated in Figure 3?

8) Can the presence or absence of radial symmetry be evaluated statistically?

9) Patient selection - I note that previous papers (e.g. Merricks et al 2021) had a larger patient cohort - is the cohort presented here a convenience sample? Or were patients subselected based on specific

criteria.

10) In the discussion, it might be worth discussing whether the authors think this method may have utility outside of the pathological dynamics observed during an epileptic seizure?

11) regarding data availability - (1) please state whether patients have provided informed consent regarding the use of their data for research use; (2) sharing of non-identifiable information such as isolated multi-electrode recordings, should not be prohibited by HIPAA privacy rule (<https://www.hhs.gov/hipaa/for-professionals/special-topics/research/index.html>). If data are not being shared for other reasons, those should be stated.

Reviewer #2 (Remarks to the Author):

The paper of Lee et al presents a novel analysis of MEA data, containing both spikes and LFPs, recorded in human patients.

This is obviously an interesting and important topic, but the manuscript did not convince me that the methodological approach is sound.

In particular the argument for why a sinc-function in time implies a sinc-function in space is unclear to me.

Is this suggested as a general result independent of data if spatiotemporal separability is assumed? And is the assumption of spatiotemporal separability warranted?

Further, the st-SCA method as it is presented seems to me to treat positions close to center of the MEA (which has many contributions to the average) differently than the positions at the edge of the MEA.

Minor points:

- The notation " $st - SCA(\xi, \psi, \tau)$ " (for example, in Equation (3)) as the hyphen can be interpreted as a minus sign.

Reviewers #3-4 (Remarks to the Author):

This paper reports data from recordings using a 10x10 (0.5 mm spacing) microelectrode array (MEA; 4x4 mm) placed on the surface of the cortex in patients with focal epilepsy. In five patients, recordings were performed in regions determined to be "recruited" during seizures, and in two patients, the recordings were performed at loci not "recruited" during seizures but presumably within the seizure onset zone (determined by unspecified clinical criteria, but presumably not displaying ictal activity). A novel method of "spatiotemporal spike-centered averaging" (st-SCA) was developed in which "spikes" (multiunit action potentials) at various electrodes in the array were used to trigger averages of potentials across the MEA, after first aligning the active electrode to the center of the array space. The st-SCA from "recruited" cortex was distinct from "un-recruited" cortex in that it was large in amplitude and formed an annular pattern surrounding the trigger electrode, with a negative center and alternating positive and negative rings over an approximately 1.5 mm spatial scale. The st-SCA from un-recruited cortex was lower in amplitude with no apparent annular pattern. It is concluded that the annular pattern of st-SCA in recruited cortex may reflect (1) an excitatory focus centered on spiking cortex, (2) surrounded at millimeter distances by inhibition, and then (3) the inhibitory "surround" is itself surrounded by synaptic excitation, which is thought to reflect "mid-range" connectivity. The authors conclude that these mid-range excitatory connections may uniquely provide the excitatory synaptic drive that underlies focal seizures. However, the manuscript in general, and this conclusion in particular, is weakened by several key points.

From the precise recording configuration under these clinical conditions, it is not clear what the underlying question or hypothesis is. If one experimental "group" has more of a "signal" than the other "group", how is this to be interpreted? Are the authors saying or concluding that the pattern of activity they describe is important in terms of how an action potential (i.e., the triggering event) causes changes in the spatio-temporal organization of the LFP? The data are relevant to the time period during a seizure, but what is the actual interpretation of the data and the present analysis for the ictal versus inter-ictal periods? Is it that single action potentials activate local excitatory circuits during seizures, and cause changes in the spatio-temporal organization of the surrounding LFPs during a seizure? But not during the inter-ictal period?

Critical issues and concerns:

- 1) The authors interpret the polarity of averaged field potentials in st-SCA to directly reflect the excitatory/inhibitory nature of underlying synaptic activity. Thus, they conclude that the annular pattern in recruited cortex indicates a center of excitation (because it is negative) surrounded by inhibition (because it is positive) surrounded by another ring of excitation (negative) mediated by mid-range excitatory connectivity. Based on these assumptions, the authors propose a candidate mechanism for seizure recruitment and propagation involving "pathological escape of uncontrolled excitation across the cortex", an "escape of hyperexcitation" (presumably beyond the surrounding ring of inhibition) via a "jump that engages mid-range connectivity in the millimeter range." Although this hypothesis is interesting and may be true, this physiological model is not justified by the current data, at least not without a better and more clear explanation. It has long been established that the polarity of field potentials recorded at the cortical surface is determined by the locus of the relevant synaptic input as well as the direction of transmembrane currents imposed predominantly on the parallel apical dendrites of pyramidal cells. The equivalent current dipole formed by these dendrites may produce a negative surface potential due to depolarization at the distal regions or, conversely, hyperpolarization proximal to the soma. Without concomitant laminar MUA recording, excitation versus inhibition for a given surface field potential cannot be determined. This seems to be a core problem with the interpretations. Furthermore, is it not possible that the opposite scenario explains the data equally well: local inhibitory circuits may include pyramidal cell connections to nearby interneurons that then form connections to nearby pyramidal cells adjacent to the initial pyramidal cell, which had the action potentials that were used as the trigger? The rationale for interpreting the data as described is hard to follow, and not necessarily valid.
- 2) Key recording details are omitted. Although other publications were sometimes cited, interpretation of the results is difficult. Presumably the MEA is implanted at a sub-dural site on the cortical surface, but this is not clearly stated. The details of the MEA by Blackrock Microsystems are not specified – does it have flat contacts or is it the "bed-of-nails" type of array? One assumes that all recordings were performed from lateral temporal cortex due to accessibility during surgery, but it is not clear how these regions (as opposed to deeper hippocampus) could be considered the seizure focus (in two "un-recruited" patients) or could be closely "recruited" during seizures in the other patients. This type of information is important for interpretation of the MUA data reported here.
- 3) It is noted that ictal STA's were recorded "during and around" ictal activity. These different recording periods should be better specified and perhaps analyzed separately, since cortical excitability (and thus all of the data reported here) could be quite different for pre-ictal, ictal and post-ictal periods (and as a function of the time before and after the seizure).
- 4) The st-SCA for some focal seizures indicated a sinc function (Fourier transform of a rectangular function) in both the spatial and temporal dimension. It is not clear why this would be the case or what could be concluded about the underlying physiology based on this observation. Similarly, it is not clear why a sinc function in the temporal domain should predict a sinc function in the spatial domain as stated. It would seem that they would be unrelated.
- 5) Please provide a better description of exactly what the "recruited" vs "un-recruited territory" is? How exactly is it defined, and how is the reader to interpret the results in the context of these two "territories"? And how is all of this related to the "seizure-onset zone." Better illustrations, with simple diagrams, would probably be useful.

6) Several other issues of interpretation should be discussed. For example, a single spike is used as the trigger, but classical studies with paired recordings by Miles and Wong showed that single spikes evoke small EPSPs, and these are rare (probably in only a few pyramidal cells), so this spike-triggered response would seem to be quite small and rare. According to this earlier work, the critical signal to activate local excitatory circuits is a burst of spikes when inhibition is depressed.

7) Another concern is that the authors are conducting these analyses on spike-triggered LFPs during seizures – although interesting, are the authors assuming that inhibition is depressed? Presumably so, but Miles and co-workers have more recently argued that GABAA-mediated “inhibition” becomes excitatory in tissue resected from epileptogenic human tissue, so how would that observation influence interpretations? Since the polarity of LFPs does not indicate excitation versus inhibition (see above), this also seems problematic.

Summary conclusion:

A core problem is that the paper is quite difficult to read and understand. Even readers with interest and expertise in the role of the electrophysiological properties of cortical neuronal circuits in seizure propagation would probably not be able to follow the logic, methods, results, and potential importance of this work. Many of the terms and concepts are not adequately explained, at least not sufficiently clear for many readers. Another problem is that the interpretations, which could be important, seem to be over-stated. The latter potential problem arises, at least in part, from the former problem. Thus, the paper should be revised to better explain all aspects of the research.

Based on the issues described above, is the paper in its present form too specialized for this journal, possibly even for the computer science and biomedical engineering component of the readership? In its present form, most readers will have a difficult time understanding the potential importance of the work.

Dear Reviewers,

We are writing to submit a revision of our original research article that was originally submitted under the title “Novel visualization of the spatiotemporal relationship between ictal spiking and LFP supports the involvement of mid-range excitatory circuits during human focal seizures.”

Based on the feedback from the reviewers, it was evident that our presentation was difficult to understand, especially the section describing the spatiotemporal relationship. Therefore, we have made several large, structural changes to the manuscript and updated its title to better reflect its contents. The manuscript has now been renamed to “Symmetry of temporal and spatial components of the spike-LFP relationship during human focal seizures.” In addition, based on the reviewer’s comments and suggestions, we have made significant modifications to the text and some figures.

We thank the reviewers for taking the time to read our manuscript and for providing thoughtful comments that we have addressed in our revision. Below is a point-by-point response to the comments from the reviewers. All changes are described below in this letter and highlighted in the main text by the comment feature.

Thank you very much for your time and consideration of this manuscript.

Somin Lee
Sarita Deshpande
Wim van Drongelen
The University of Chicago
Chicago, IL, USA

Responses to Reviewer 1

Major comments

1A. How does the suggested method differ from other approaches that translate spatiotemporal patterns into 3D maps (e.g. statistic parametric mapping performed on ERP data - e.g. Stefanics et al. 2018 <https://www.jneurosci.org/content/38/16/4020/tab-figures-data> - Fig 3) - I'm assuming the answer here lies in the ability to generate maps for several concurrently occurring spiking patterns? Could the same be achieved by generating statistic parametric maps on transformed coordinates for each spike location? Would these two approaches be equivalent? How can the authors' approach be used to make statistical inference regarding the observations they have described in the paper?

Our method is in principle the same as computing a spatial potential distribution for an evoked/event-related potential. However, in these procedures, the trigger/stimulus is well defined and precisely repeated at the same location. The difference is that in our case, the spatiotemporal average depends on a trigger that isn't constrained spatially because an ictal action potential can occur across the spatial domain of the MEA. The text in the Supplemental Materials has been modified to clarify this point (Supplemental Materials, page 5, lines 104-105) and is repeated below for convenience:

“This method to determine spatiotemporal patterns is based on a spike trigger that is not constrained spatially because an ictal action potential can occur across the spatial dimension of the MEA.”

We show the statistical significance of our findings by showing the noise estimates for both the spatial and temporal results (Fig. S4, page 10, lines 150-159). For these estimates, we employ the so-called plus-minus average calculation (van Drongelen, 2018; Section 4.4.3).

Interpretation of these results is now described in the Discussion section (page 10, lines 310-314). The text is repeated below for convenience:

“A natural and necessary question to ask at this junction is whether the 2D sinc function in the spatial domain has any biological significance. While the exact mechanisms underlying temporal and spatial sinc patterns are beyond the scope of this study, we propose here that the concentric “donut-ring” pattern in the spatial component of the spike-LFP relationship may reflect the engagement of mid-range horizontal connections during seizure initiation and propagation.”

Minor comments

1B. The reference in line 72 as given refers to a whole book - it would be helpful to refer the reader to the specific section/chapter that supports the statement made

This reference has been switched to the one listed below, which is a chapter of different textbook. This references includes specifically the rectangle function and sinc function in their list of Fourier transforms.

Boashash, B. (2016). Chapter I: The Time-Frequency Approach: Essence and Terminology. In B. Boashash (Ed.), Time-Frequency Signal Analysis and Processing (Second Edition) (pp. 3-29). Academic Press. <https://doi.org/https://doi.org/10.1016/B978-0-12-398499-9.09991-X>

1C. It would be useful to define 'mesoscale' in spatial terms as used here for the reader - I see these terms are later mapped in Figure 6, but this would be helpful earlier in the manuscript text

The following statement indicating that "mesoscale" refers to activity in the millimeter range has been added in the Introduction (page 3, 61-64) and is repeated below for convenience:

"For example, large-scale (cm-range) global activity can be recorded by macroelectrodes at the scalp or cortex, and mesoscale (mm-range) and microscale (sub-mm range) activity can be recorded by intracranial arrays or bundles of microelectrodes (Eissa et al., 2017; Eissa et al., 2016; Schevon et al., 2012)."

1D. Figure 3 - it is unclear to me what the z-axis is on this plot? xy are in mm so presumably spatial - does z represent uV of the signal that is also colour coded?

Yes, the reviewer is correct that the z-axis in panels E and F of this figure represent uV that is also represented by the color scale. The legend has been updated to specifically explain that the z-axis in this topographical view is the uV and redundant with the color scale (Figure 4; page 20, lines 529-530) and is repeated below for convenience:

"The ξ -axis and ψ -axis represent the spatial dimensions of the MEA, and the third dimension (z-axis) in this topological view represents microvolt (μ V) units. The grayscale corresponds to the z-axis and is in μ V units."

Note: This figure is now Figure 4 in the revision.

1E. Providing an example plot of the sinc function would be useful

A figure has been added to the supplementary materials (Figure S1) that illustrate the sinc function in both 1 dimension (D) and 2 dimensions (Supplemental Materials; page 7, lines 128-130). In addition, newly added Fig. 6Ai and 6Aii include examples of 1D and 2D sinc functions (pages 22-23, lines 545-552).

1F. In the discussion, the authors mention that the secondary generalisation may contribute to differences in observation - did the authors use the whole seizure, or segments of the seizure for their analysis? If the latter, it would be useful how these relate to time of generalisation?

For the current study, the entirety of the seizure was included in the analyses. This fact has been made clearer by the following edits to the Methods section (lines 377-378) and is repeated below for convenience:

"The entirety of the ictal segments and two-minute interictal segments were used in these analyses (Table S1)."

1G. Figure 6A - it is not quite clear what the different cells / dendritic trees are / what the horizontal line represents? More labels could be useful

The histology figure has now been updated with labels 'E' for excitatory neuron and 'I' for inhibitory neuron. In addition, the three scales of connectivity are denoted as the three arrows: sub-mm connectivity, mm-range connectivity (marked in red), and cm-range connectivity via white

matter tracts (Figure 6; page 22-23, lines 557-565). This is now explicitly stated in the legend and is repeated below for convenience:

“Left panel: Diagram of gray matter excitatory connections of a neocortical pyramidal cell showing the short-range connections (order of 100s of μm) and mid-range connections (order of mm, labeled in red) via the pyramidal cell axon collaterals (based on Fig. 5 in Nieuwenhuys, 1994). Excitatory neurons are labeled “E,” and the inhibitory neuron is labeled “I” in this schematic.”

Note: Figure 6A is now Figure 6C in the revision.

1H. Could a representative trace of the LFP signal be included for the two patients illustrated in Figure 3?

Representative LFP traces showing the average activity across the MEA for Patient 1 and Patient 5 have now been added to this figure (Figure 4, Panels A-B; page 20, lines 522-523).

Note: Figure 3 is now Figure 4 in the revision.

1I. Can the presence or absence of radial symmetry be evaluated statistically?

Symmetry could be quantified by computing a metric that describes the variance between several “slices” of the circular shape. While this would allow the symmetry to be quantified, the determination of the “presence” of symmetry would involve an arbitrary threshold in the individual observations. Consequently, we offer the radial symmetry as a qualitative observation in this current study. If studies on larger datasets are made possible in the future, certainly, a more specific metric would be warranted and appropriate. The text has been modified in the Results section (page 9, lines 274-276) and is repeated below for convenience:

“Taking advantage of the qualitatively observed radial symmetry observed in the st-SCA, we converted the Cartesian coordinates (ξ, ψ) into polar coordinates (r, θ) and focused on the spatial relationship with respect to r (Fig. 5A).”

1J. Patient selection - I note that previous papers (e.g. Merricks et al 2021) had a larger patient cohort - is the cohort presented here a convenience sample? Or were patients subselected based on specific criteria.

Previous papers such as Merricks, et al. (2021) have a larger patient cohort because they include both Utah microelectrode array data and Behnke-Fried depth electrode data. Behnke-Fried data is more abundant, but it does not allow detailed characterization of spatial patterns as these probes generally have only 8 microelectrode tips that spray out in an irregular fan pattern. We only utilized Utah microelectrode array data which accounts for the smaller cohort. The text in the Methods section (page 13, lines 385-390) was updated and is repeated below for convenience:

“A 96-channel, 4 x 4mm MEA (Utah array; Blackrock Microsystems) was implanted into neocortical gyri along with subdural electrodes (ECoG). The 96 microelectrodes were 1mm in length and arranged in a regular 10x10 grid pattern with empty corners. The location of the MEA implantation was the clinically determined seizure onset zone (SOZ) that was characterized prior to implantation. Expanded and complete details of study enrollment, clinical evaluation of the SOZ, surgical procedures, and recording parameters have been previously published (Schevon et al., 2012; Truccolo et al., 2014).”

1K. In the discussion, it might be worth discussing whether the authors think this method may have utility outside of the pathological dynamics observed during an epileptic seizure?

Indeed, we believe that our st-SCA method can be used for a broad scope of applications. The following text has been added (repeated below for convenience) to the Discussion section (lines 366-371) to include applications beyond epileptic pathology:

“Furthermore, because this novel method includes the spatial domain, the st-SCA method can be used for a broad scope of applications, such as MEA cultures (Cotterill et al., 2016; Kapucu et al., 2022), Utah arrays implanted in monkeys completing tasks (Brochier et al., 2018; Dickey et al., 2009; Manyakov & Van Hulle, 2010), MEAs implanted in humans for sleep (Le Van Quyen et al., 2016), and for brain-computer interfaces (Maynard et al., 1997; Woepfel et al., 2021).”

In addition, we discuss the applications of our method to recording modalities with lower spatial resolution in comparison to 10x10 Utah arrays, such as Behnke-Fried electrodes in lines 298-308 of the Discussion section and Figure 6B:

“This predictive power is important in the context of clinical microelectrode recordings because it suggests that it may be possible to characterize spatial patterns without the use of gridded MEAs. While MEAs are advantageous for monitoring and studying seizure activity with high temporal and spatial resolution, their current clinical utility is limited as they cannot be easily used to sample from multiple cortical areas. Interestingly, we found that the sinc function can be characterized in the temporal domain by using spiking and LFP information from a random subset of only eight electrodes (Fig. 6B). Although the spatial pattern is impossible to discern with just eight electrodes, the underlying spatial pattern may be inferred to be a 2D sinc since the associated temporal pattern is a sinc function. This suggests that the st-SCA may be characterized by using neocortical microelectrodes that allow for recording from multiple areas by reducing the number of channels per probe. The development of such electrodes is technologically feasible as similar probes already used clinically for the monitoring of deep brain structures such as Behnke-Fried depth electrodes (Misra et al., 2014).”

1L. Regarding data availability - (1) please state whether patients have provided informed consent regarding the use of their data for research use; (2) sharing of non-identifiable information such as isolated multi-electrode recordings, should not be prohibited by HIPAA privacy rule (<https://www.hhs.gov/hipaa/for-professionals/special-topics/research/index.html>). If data are not being shared for other reasons, those should be stated.

(1) A statement confirming that informed consent was obtained has been added to the Methods section under the subheading “Patients” (page 13; lines 381-382) and is repeated below for convenience:

“All patients provided informed consent regarding the use of their data for research purposes.”

(2) Upon investigation into the history of this dataset, we found that open data sharing is limited not by HIPAA but by IRB protocols that restrict transferring this data between labs and institutions. The data availability statement has been amended to reflect this (page 15, lines 440-444) and is repeated below for convenience:

“The data used in this study is not publicly available due to IRB protocols at the University of Chicago, Columbia University, and Harvard University and all data sharing will have to follow protocols compliant with registered IRB protocols. Questions about data availability can be addressed at the corresponding author (W.v.D).”

Responses to Reviewer 2

Reviewer #2 (Remarks to the Author):

The paper of Lee et al presents a novel analysis of MEA data, containing both spikes and LFPs, recorded in human patients.

This is obviously an interesting and important topic, but the manuscript did not convince me that the methodological approach is sound.

Major comments

2A. In particular, the argument for why a sinc-function in time implies a sinc-function in space is unclear to me. Is this suggested as a general result independent of data if spatiotemporal separability is assumed? And is the assumption of spatiotemporal separability warranted?

This comment was made in some form by all reviewers, which motivated us to re-arrange the presentation in our manuscript to highlight the significance of the mathematical derivations we present and how they link to our MEA analyses. The symmetry between the temporal and spatial domains is a consequence of mathematical derivation that is independent from any data. Analysis of the microelectrode array data allows us to comment that this symmetry is not simply a mathematical oddity, but a feature that manifests in real world EEG recordings.

We have made the following updates to the manuscript in order to clarify this important result:

- (1) The mathematical derivation portion has been moved to the beginning of the Results section to separate it from any analysis of real EEG recordings (Results: “Theoretical model reveals symmetry between the temporal and spatial components of the spike-LFP relationship”; pages 5-6, lines 102-152).*
- (2) Explanation of the steps involved in the mathematical derivation have been expanded to include more step-wise detail (Eqs. 1-6; pages 5-6, lines 117-145).*
- (3) A supplementary text section that describes the mechanics of Fourier transforms and Fourier pairs has been added. This text also refers to a textbook chapter for those who would like to learn more about the mathematics of Fourier transforms (Supplemental Materials, page 2, lines 8-24).*
- (4) A supplementary figure (Supplemental Materials; Figure S1, page 7, lines 126-130) has been added to illustrate sinc functions in one and two dimensions (D). In addition, Panels Ai and Aii have been added to Figure 6 depicting simulated 1D and 2D sinc functions (pages 22-23, lines 527-532).*

(5) We have made edits throughout the text (namely the Results and Discussion sections) to emphasize the fact that the analysis of microelectrode array recordings serves to demonstrate that this mathematical symmetry can be observed in real EEG recordings. Figure 6A has been newly added to highlight this mathematical symmetry (pages 22-23, lines 525-545).

(6) Figure 6 has been expanded and reorganized to help summarize the major findings and discussion points of our manuscript. Through this figure and accompanying text edits, we discuss the potential biological implications of finding a 1D sinc function in the temporal domain and a 2D sinc function in the spatial domain (pages 22-23, lines 525-545).

2B. Further, the st-SCA method as it is presented seems to me to treat positions close to center of the MEA (which has many contributions to the average) differently than the positions at the edge of the MEA.

We are interested in individual spike-LFP relationship. We acknowledge that because there are more spikes near the center of the MEA, there are more contributions to the average. However, we also normalize the contributions by the number of spikes in our analyses (Supplemental Materials, page 5, lines 81-105) and check for a strong signal-to-noise ratio of these contributions (Supplemental Materials, Fig. S4, page 10, lines 150-159).

Minor comments

2C. The notation " $st - SCA(\xi, \psi, \tau)$ " (for example, in Equation (3)) as the hyphen can be interpreted as a minus sign.

This has been fixed throughout the manuscript.

Responses to Reviewers 3 & 4

Reviewers #3-4 (Remarks to the Author):

This paper reports data from recordings using a 10x10 (0.5 mm spacing) microelectrode array (MEA; 4x4 mm) placed on the surface of the cortex in patients with focal epilepsy. In five patients, recordings were performed in regions determined to be "recruited" during seizures, and in two patients, the recordings were performed at loci not "recruited" during seizures but presumably within the seizure onset zone (determined by unspecified clinical criteria, but presumably not displaying ictal activity). A novel method of "spatiotemporal spike-centered averaging" (st-SCA) was developed in which "spikes" (multiunit action potentials) at various electrodes in the array were used to trigger averages of potentials across the MEA, after first aligning the active electrode to the center of the array space. The st-SCA from "recruited" cortex was distinct from "un-recruited" cortex in that it was large in amplitude and formed an annular pattern surrounding the trigger electrode, with a negative center and alternating positive and negative rings over an approximately 1.5 mm spatial scale. The st-SCA from un-recruited cortex was lower in amplitude with no apparent annular pattern. It is concluded that the annular pattern of st-SCA in recruited cortex may reflect (1) an excitatory focus centered on spiking cortex, (2) surrounded at millimeter distances by inhibition, and then (3) the inhibitory "surround" is itself surrounded by synaptic excitation, which is thought to reflect "mid-range" connectivity. The authors conclude that these mid-range excitatory connections may uniquely provide the excitatory synaptic drive that underlies focal seizures. However, the manuscript in general, and this conclusion in particular, is weakened by several key points.

3A. From the precise recording configuration under these clinical conditions, it is not clear what the underlying question or hypothesis is.

Our hypotheses are that spatiotemporal relationships between the spike and LFP may be governed by:

- (1) A special, mathematically derived symmetry associated with the sinc function. (Fig. 1, Fig. 6A, Eq. 4-6)*
- (2) The overall network state (ictal vs. non-ictal) (Fig. 3, Fig. S3)*
- (3) The location of the network (recruited vs. non-recruited seizure territories) (Fig. 3, Fig. 4, Fig. S2, Fig. S3)*

We have added the following text to the Introduction section (page 3, lines 86-88) to make these hypotheses more explicit:

“We hypothesize that this spatiotemporal relationship is governed by the network state (ictal vs. non-ictal) and the location in the network (recruited vs. unrecruited seizure territory).”

3B. If one experimental “group” has more of a “signal” than the other “group”, how is this to be interpreted? Are the authors saying or concluding that the pattern of activity they describe is important in terms of how an action potential (i.e., the triggering event) causes changes in the spatio-temporal organization of the LFP?

For our studies, we employ the plus-minus average to show that while there is significant signal in ictal states (note that we find strong signals – application of the rigorous so-called five sigma rule translates into a signal to noise ratio above 14dB and our findings are well above that level, e.g. Fig. S4). This comment is similar to one brought up by Reviewer 1 (Response 1A). Our Response 1A is repeated below for convenience:

“We show the statistical significance of our findings by showing the noise estimates for both the spatial and temporal results (Fig. S4). For these estimates, we employ the so-called plus-minus average calculation (van Drongelen, 2018; Section 4.4.3).”

In addition, we repeat below for convenience the modified caption for Fig. S4 (Supplemental Materials, page 10, lines 150-159):

“Noise estimates of the spatiotemporal spike-centered average (st-SCA) of Patient 1. The SNR of the signals is well above 14dB in each panel (as per application of the so-called five sigma rule).”

Interpretation of these results is now described in the Discussion section (page 10, lines 310-314). The text is repeated below for convenience:

“A natural and necessary question to ask at this junction is whether the 2D sinc function in the spatial domain has any biological significance. While the exact mechanisms underlying temporal and spatial sinc patterns are beyond the scope of this study, we propose here that the concentric “donut-ring” pattern in the spatial component of the spike-LFP relationship may reflect the engagement of mid-range horizontal connections during seizure initiation and propagation.”

3C. The data are relevant to the time period during a seizure, but what is the actual interpretation of the data and the present analysis for the ictal versus inter-ictal periods? Is it that single action potentials

activate local excitatory circuits during seizures, and cause changes in the spatio-temporal organization of the surrounding LFPs during a seizure? But not during the inter-ictal period?

Our hypotheses (as listed in Response 3A) are that the spike-LFP relationship changes with network state (ictal vs. non-ictal) and the location of the network (recruited vs. non-recruited seizure territories).

For example, the temporal spike triggered average (STA) only shows strong signal in recordings obtained from MEAs implanted in recruited seizure territory (Fig. S2A-E), while there is much smaller amplitude signals in STA calculations obtained from MEAs implanted in non-recruited territories (Fig. S2F, G). This carries over into the spatial realm as a “donut” or “sinkhole” shape is observed in recordings from recruited territories (Fig. S2A-E), while there is no such organization in the spatial domain for recordings from unrecruited territories (Fig. S2F, G).

This result applies when comparing ictal states. There is much stronger signal from recordings obtained during seizures (Fig. S3A, C, E), while there is weaker signal during interictal states (Fig. S3B, D, F).

An example of the noise estimate procedure is depicted in Fig. S4.

All the calculations performed for this study are correlative, which means that we can say that action potentials are associated with significant changes in spatiotemporal organizations based on network state or location. It is difficult to say that the action potentials “cause” these changes in spatiotemporal organization. (Note that the sinc function that fits so well with part of the ictal observations is an example of a noncausal case). Our interpretation is that the spike-LFP relationship during seizures is dominated by net excitation. Patterns that resemble a “donut” in the spatial domain where excitation is observed in rings separated by ~2.5mm may reflect the engagement of mid-range connections.

This interpretation can be found in the Discussion section (pages 10-11, lines 321-336) and is repeated below for convenience:

“In our discussion of the biological implications of the observed st-SCAs, we adopt the interpretation for our specific electrode configuration as previously described (Eissa et al., 2017) by assigning a net excitation to negative deflections and net inhibition to positive deflections. This interpretation is also in line with previous studies of the ictal core and propagation (Schevon et al., 2012; Tryba et al., 2019). Accordingly, our st-SCA analyses (Fig. 3, 4) show that in the recruited ictal territory, the spike-LFP correlation at small lags is dominated by net excitation during seizures in all patients. The activity level in the excitatory center, representing the activity at the ictal wave, is excessively high, possibly due to saturation of the local inhibitory population (Tryba et al., 2019). In Patients 1-3 we also observe a ring of reduced excitation at a distance ~1.5mm around the excitatory center (Fig. 4E, S2A-C). In turn, the ring of reduced excitation is surrounded by a second ring at an additional distance of ~1mm where excitation increases again. For these patients, this donut-shaped st-SCA is specific to the recruited seizure territory in the ictal phase (Fig. S2A-C, S3A). This observation suggests that the ictal wave in the recruited territory, represented by the excitatory center ($\xi, \psi=0,0$), creates an escape of hyperexcitation via a jump that engages mid-range connectivity in the millimeter range. Decorrelation of the LFP prior to the st-SCA calculations yielded similar spatiotemporal patterns (Fig. S7), further corroborating the importance of local millimeter range excitatory connections in focal seizures.”

Critical issues and concerns:

3D. The authors interpret the polarity of averaged field potentials in st-SCA to directly reflect the excitatory/inhibitory nature of underlying synaptic activity. Thus, they conclude that the annular pattern in recruited cortex indicates a center of excitation (because it is negative) surrounded by inhibition (because it is positive) surrounded by another ring of excitation (negative) mediated by mid-range excitatory connectivity. Based on these assumptions, the authors propose a candidate mechanism for seizure recruitment and propagation involving “pathological escape of uncontrolled excitation across the cortex”, an “escape of hyperexcitation” (presumably beyond the surrounding ring of inhibition) via a “jump that engages mid-range connectivity in the millimeter range.” Although this hypothesis is interesting and may be true, this physiological model is not justified by the current data, at least not without a better and more clear explanation. It has long been

established that the polarity of field potentials recorded at the cortical surface is determined by the locus of the relevant synaptic input as well as the direction of transmembrane currents imposed predominantly on the parallel apical dendrites of pyramidal cells. The equivalent current dipole formed by these dendrites may produce a negative surface potential due to depolarization at the distal regions or, conversely, hyperpolarization proximal to the soma. Without concomitant laminar MUA recording, excitation versus inhibition for a given surface field potential cannot be determined. This seems to be a core problem with the interpretations. Furthermore, is it not possible that the opposite scenario explains the data equally well: local inhibitory circuits may include pyramidal cell connections to nearby interneurons that then form connections to nearby pyramidal cells adjacent to the initial pyramidal cell, which had the action potentials that were used as the trigger? The rationale for interpreting the data as described is hard to follow, and not necessarily valid.

The reviewers are correct in that alternatives are possible. As is often the case in neuroscience, the degrees of freedom in evaluating findings allow alternative interpretations. In this current study, we employ the interpretation that is consistent with other studies that have been published with this data set (Eissa et al., 2017). This interpretation is corroborated by both and clinical electrophysiology studies of ictal core dynamics (Schevon et al., 2012; Tryba et al., 2019). Many of these studies also use the same or similar datasets as the one used for this study. We have added the following text to the Discussion section (pages 10-11, lines 321-325) to clarify these points and to explicitly state our assumptions:

“In our discussion of the biological implications of the observed st-SCAs, we adopt the interpretation for our specific electrode configuration as previously described (Eissa et al., 2017) by assigning a net excitation to negative deflections and net inhibition to positive deflections. This interpretation is also in line with previous studies of the ictal core and propagation (Schevon et al., 2012; Tryba et al., 2019).”

3E. Key recording details are omitted. Although other publications were sometimes cited, interpretation of the results is difficult. Presumably the MEA is implanted at a sub-dural site on the cortical surface, but this is not clearly stated. The details of the MEA by Blackrock Microsystems are not specified – does it have flat contacts or is it the “bed-of-nails” type of array? One assumes that all recordings were performed from lateral temporal cortex due to accessibility during surgery, but it is not clear how these regions (as opposed to deeper hippocampus) could be considered the seizure focus (in two “un-recruited” patients) or could be closely “recruited” during seizures in the other patients. This type of information is important for interpretation of the MUA data reported here.

To clarify, MEAs for this study were implanted in neocortex, not hippocampus. All details with respect to recording parameters are described in several published studies (Merricks et al., 2021; Schevon et al., 2012; Truccolo et al., 2014). 4x4mm MEAs were implanted in cortex involved in seizures. The microelectrodes all had a length of 1mm and their position in neocortex was confirmed histologically (Schevon et al., 2012). Recordings were obtained during the presurgical evaluation of the patients. While we cannot repeat all previously published details in our manuscript, we have expanded the summarized methods in our manuscript to include more details:

As a point of clarification, the clinically defined seizure onset zone (SOZ) is not equivalent to recruited cortex. The SOZ is a clinical definition, whereas recruited territory is a concept that was developed in the process of basic science research performed with this data. Whether a territory is recruited or not can only be determined after MEA implantation by correlating spiking activity to global seizure rhythms (Schevon et al., 2012; Eissa et al., 2017; Merricks et al., 2021). The SOZ is a clinical concept and is defined as the electrodes that first observe seizure activity. One of the most significant findings that has arisen from this dataset is that an area broadly characterized as the SOZ during clinical evaluation could be further subdivided into different territories (recruited vs. unrecruited). We have added more explicitly to the methods section that the MEAs were implanted in the clinically determined SOZ (page 13, lines 387-387) and is repeated below for convenience:

“The location of the MEA implantation was the clinically determined seizure onset zone (SOZ) that was characterized prior to implantation.”

We have also added Panel A to Figure 2 (page 17-18, lines 469-473) in the manuscript describing the definitions of recruited vs. unrecruited territory in more detail.

We hope that these additions will further clarify how the location of the MEAs are determined.

3F. It is noted that ictal STA's were recorded “during and around” ictal activity. These different recording periods should be better specified and perhaps analyzed separately, since cortical excitability (and thus all of the data reported here) could be quite different for pre-ictal, ictal and post-ictal periods (and as a function of the time before and after the seizure).

We computed the st-SCAs separately for ictal (seizure) segments and interictal (between seizure) segments. The phrase “during and around” has been removed and clarified. Table S1 denotes the duration and number of spikes per segment per patient. Interictal activity is defined under the section “Visualization of spatial and temporal components of the spike-LFP relationship” (page 8, lines 227-228) and is repeated below for convenience:

“...interictal was defined as being at least two hours away from any known ictal activity.”

3G. The st-SCA for some focal seizures indicated a sinc function (Fourier transform of a rectangular function) in both the spatial and temporal dimension. It is not clear why this would be the case or what could be concluded about the underlying physiology based on this observation. Similarly, it is not clear why a sinc function in the temporal domain should predict a sinc function in the spatial domain as stated. It would seem that they would be unrelated.

This comment was made in some form by all reviewers, which motivated us to re-arrange the presentation in our manuscript to highlight the significance of the mathematical derivations we present and how they link to our MEA analyses. The symmetry between the temporal and spatial domains is a consequence of mathematical derivation that is independent from any data. Analysis of the microelectrode array data allows us to comment that this symmetry is not simply a mathematical oddity, but a feature that manifests in real world EEG recordings.

We have made the following updates to the manuscript in order to clarify this important result:

- (1) The mathematical derivation portion has been moved to the beginning of the Results section to separate it from any analysis of real EEG recordings (Results: “Theoretical model reveals symmetry between the temporal and spatial components of the spike-LFP relationship”; pages 5-6, lines 102-152).*
- (2) Explanation of the steps involved in the mathematical derivation have been expanded to include more step-wise detail (Eq. 1-6; pages 5-6, lines 117-145).*
- (3) A supplementary text section that briefly describes the mechanics of Fourier transforms and Fourier pairs has been added. This text also refers to a textbook chapter for those who would like to learn more about the mathematics of Fourier transforms (Supplemental Materials, page 2, lines 8-24).*
- (4) A supplementary figure (Figure S1; page 7, lines 126-130) has been added to illustrate sinc functions in one and two dimensions (D). In addition, Figure 6 now includes Panels Ai and Aii as examples of sinc functions in 1D and 2D (page 22-23, lines 524-544).*
- (5) We have made edits throughout the text (namely the Results and Discussion sections) to emphasize the fact that the analysis of microelectrode array recordings serves to demonstrate that this mathematical symmetry can be observed in real EEG recordings. Figure 6A has been newly added to highlight this mathematical symmetry (pages 22-23, lines 524-544).*
- (6) Figure 6 has been expanded and reorganized to help summarize the major findings and discussion points of our manuscript. Through this figure and accompanying text edits, we discuss the potential biological implications of finding a 1D sinc function in the temporal domain and a 2D sinc function in the spatial domain.*

Note: This response is the same as Response 2A since Review 2 shared the same concern. The response has been repeated here for the convenience of Reviewers 3 and 4.

3H. Please provide a better description of exactly what the “recruited” vs “un-recruited territory” is? How exactly is it defined, and how is the reader to interpret the results in the context of these two “territories”? And how is all of this related to the “seizure-onset zone.” Better illustrations, with simple diagrams, would probably be useful.

The definitions of recruited and unrecruited territories have been expanded under the “Visualization of spatial and temporal components of the spike-LFP relationship” section of the results (page 8, lines 222-226) and is copied below for convenience:

“As previously described (Schevon et al., 2012; Truccolo et al., 2011), recruited seizure territory is an area of tissue that is invaded by the ictal wavefront throughout the course of a seizure. The ictal wavefront is defined by high rates of firing that is highly correlated with overlying low frequency rhythms. Unrecruited territory sees no invasion of the ictal wavefront but still shows rhythmic EEG activity due to local synaptic activity (Merricks et al., 2021; Schevon et al., 2012).”

Additionally, a figure panel (Fig. 2A; pages 17-18, lines 468-472) has been added to help visually explain the definitions of recruited and unrecruited territory and how these territories relate to the location of the MEA. The newly added legend for Panel A is repeated below for convenience:

“Recruited territory involves a seizure passing through and invading the local cortical tissue, and unrecruited territory is tissue outside the recruited territory but may still be characterized by strong, local synaptic activity (Schevon et al., 2012; Merricks et al., 2021).”

3I. Several other issues of interpretation should be discussed. For example, a single spike is used as the trigger, but classical studies with paired recordings by Miles and Wong showed that single spikes evoke small EPSPs, and these are rare (probably in only a few pyramidal cells), so this spike-triggered response would seem to be quite small and rare. According to this earlier work, the critical signal to activate local excitatory circuits is a burst of spikes when inhibition is depressed.

Our main interest (as outlined and clarified in Response 3A, 3B, and 3C) is changes in the spike-LFP relationship according to network state (ictal vs. non-ictal) and seizure territory (recruited vs. unrecruited). Indeed, spikes are associated with quite small responses in interictal states and in recordings outside of recruited seizure territory (Fig. 3, Fig. S3). In contrast, as outlined above in reply 3B, spike-LFP correlations in ictal states and in recruited seizure territory are not small or rare (Fig. 3, Fig. S3)

3J. Another concern is that the authors are conducting these analyses on spike-triggered LFPs during seizures – although interesting, are the authors assuming that inhibition is depressed? Presumably so, but Miles and co-workers have more recently argued that GABAA-mediated “inhibition” becomes excitatory in tissue resected from epileptogenic human tissue, so how would that observation influence interpretations? Since the polarity of LFPs does not indicate excitation versus inhibition (see above), this also seems problematic.

Since our current study utilizes the same dataset as these previous studies, we have chosen to adopt the same interpretation as these other published studies. Specifically, Eissa et al. 2017 addresses in significant detail the rationale for the interpretation of polarity. As this is already published text, we do not find it appropriate to repeat in the main text of our current manuscript. It may be useful as a reference for readers in the supplementary information, but the appropriateness of including text from a paper published elsewhere is a matter of editorial judgement, and we will defer to the editors for guidance on this front.

This comment is also similar to the concerns addressed in Response 3D - we repeat our reply for convenience. The reviewers are correct in that alternatives are possible. As is often the case in neuroscience, the degrees of freedom in evaluating findings allow alternative interpretations. In this current study, we employ the interpretation that is consistent with other studies that have been published with this data set (Eissa et al., 2017). This interpretation is corroborated by both and clinical electrophysiology studies of ictal core dynamics (Schevon et al., 2012; Tryba et al., 2019). Many of these studies also use the same or similar datasets as the one used for this study.

We have added the following text to the Discussion section (pages 11, lines 321-325) to clarify these points and to explicitly state our assumptions. The parts of the manuscript addressing these aspects are repeated below for convenience.

“In our discussion of the biological implications of the observed st-SCAs, we adopt the interpretation for our specific electrode configuration as previously described (Eissa et al., 2017) by assigning a net excitation to negative deflections and net inhibition to positive deflections. This interpretation is also in line with previous studies of the ictal core and propagation (Schevon et al., 2012; Tryba et al., 2019).”

Summary conclusion:

A core problem is that the paper is quite difficult to read and understand. Even readers with interest and expertise in the role of the electrophysiological properties of cortical neuronal circuits in seizure propagation would probably not be able to follow the logic, methods, results, and potential importance of this work. Many of the terms and concepts are not adequately explained, at least not sufficiently clear for many readers. Another problem is that the interpretations, which could be important, seem to be overstated. The latter potential problem arises, at least in part, from the former problem. Thus, the paper should be revised to better explain all aspects of the research.

Based on the issues described above, is the paper in its present form too specialized for this journal, possibly even for the computer science and biomedical engineering component of the readership? In its present form, most readers will have a difficult time understanding the potential importance of the work.

Significant changes have been made to the manuscript, including clarifying definitions in the text, better diagrams, updated figures, and structural changes to the presentation of the findings. We hope and anticipate that these changes now facilitate the understanding of the paper.

We respectfully disagree with the notion that this paper is too specialized for this journal. The primary takeaway from this paper is the development of the novel st-SCA method (which accounts for spatial information in the calculation), which can be applied in a variety of settings. The various applications are discussed in text added to the Discussion section (page 12, lines 366-371):

“Furthermore, because this novel method includes the spatial domain, the st-SCA method can be used for a broad scope of applications, such as MEA cultures well plates (e.g. Cotterill et al., 2016; Kapucu et al., 2022), Utah arrays implanted in monkeys completing tasks (e.g. Dickey et al., 2009; Manyakov and Van Hulle, 2010; Brochier et al., 2018), or MEAs implanted in humans for sleep (Van Quyen et al., 2016), for brain-computer interfaces (Maynard et al., 1997; Woeppel et al., 2021).”

In addition, we discuss the applications of our method to recording modalities with lower spatial resolution in comparison to 10x10 Utah arrays, such as Behnke-Fried electrodes the Discussion section (page 10, lines 298-309) and Figure 6B:

“This predictive power is important in the context of clinical microelectrode recordings because it suggests that it may be possible to characterize spatial patterns without the use of gridded MEAs. While MEAs are advantageous for monitoring and studying seizure activity with high temporal and spatial resolution, their current clinical utility is limited as they cannot be easily used to sample from multiple cortical areas. Interestingly, we found that the sinc function can be characterized in

the temporal domain by using spiking and LFP information from a random subset of only eight electrodes (Fig. 6B). Although the spatial pattern is impossible to discern with just eight electrodes, the underlying spatial pattern may be inferred to be a 2D sinc since the associated temporal pattern is a sinc function. This suggests that the st-SCA may be characterized by using neocortical microelectrodes that allow for recording from multiple areas by reducing the number of channels per probe. The development of such electrodes is technologically feasible as similar probes already used clinically for the monitoring of deep brain structures such as Behnke-Fried depth electrodes (Misra et al., 2014).”

Reviewers' comments:

Reviewer #3 (Remarks to the Author):

This revised manuscript is greatly improved. The authors have addressed the concerns, and the revisions clarify many issues.

One concern relates to the degree that extracellular negativity always corresponds to net synaptic excitation, and vice versa with the positivity always corresponding to synaptic inhibition. The authors have attempted to address the concern with reference to their earlier work.

There is still a concern about whether the data analyses and conclusions will be fully comprehensible to a wide audience, but the authors have made a significant and good-faith attempt to address these concerns.

Reviewer #4 (Remarks to the Author):

This is a revised and improved manuscript. The authors have provided detailed and extensive responses to original critique with corresponding changes to the text and figures. Questions were raised in original review regarding the interpretation of field potential polarity recorded at the cortical surface in regards to underlying excitation and inhibition. These questions remain. Negative field potentials are assumed to reflect net excitation and positive potentials, inhibition. This is certainly a possible and plausible interpretation but is by no means indicated by the data. The authors refer to a paper by Eissa et al. 2017, as justification for the interpretation. This paper includes good discussion of dipole modeling as it relates to surface potentials, and more importantly, relates increased MUA in these specific data to field potential negativity. Thus, as noted by the authors, there is support for their interpretations of excitation and inhibition in the specific context of these data. Yet, even with extensive and thoughtful edits for clarity, the paper remains highly specialized. While this could limit the audience, the results should be interesting and important to those with requisite expertise.

Reviewer #5 (Remarks to the Author):

As I am not the one who raised the original points it is a bit hard for me to judge the response to other people's comments (I assume the original reviewer #1 was not available anymore?) but from my point of view the authors have responded to all these points in a very detailed and comprehensive manner. The same holds for the responses to the other three reviewers but again I do not think I should be the one to judge this.

But as an independent voice I would say that the revised paper (I haven't seen the original paper) has become quite clear and understandable and from my point of view deserves to be published in Communications Biology.

However, as I also did my own reading of this revised version I do have a couple of (mostly minor) additional comments:

Very first sentence: maybe "can reflect" (not only, they also reflect non-pathological activity). Or rephrase somehow.

Fig. 2: It is quite obvious that after translation to the origin the grid locations close to the center have a much higher support than those at the edges (it might be useful to get to see a support matrix for a typical example). This will lead to different error bars for the average and maybe even different expectation values (following a radial dependence) and the question comes to mind whether this can

lead to statistical irregularities, in particular since later data are transformed to polar coordinates where the r-dependence is investigated specifically.

Would it be possible to use simulations to somehow check the validity of the results (how much of the radial symmetry could be due to this effect), e.g. by assuming a constant number of spikes per pixel (with uniform spatiotemporal data and some controlled randomization) and observing the resulting r-dependence. A similar kind of "surrogate" analysis is performed in Fig. 6B.

Figs. 3, 4, and 6: Why is always patient 1 used (for example the first plot in both of these Figures is actually identical)? If results hold in a more general manner it should be possible to include more variation there (or if you would like to be consistent there show the results for the other patients (the ones relevant here) in the SM. The way it is now it somehow gives the impression that P1 is the model patient and for the others things do not work out as well (and I mean even the others that show temporal and spatial patterns consistent with a symmetric relationship).

References:

Many references seem to be missing, e.g. all the ones cited at the end of the Discussion (e.g Cotterill et al., 2016; Kapucu et al., 2022; Le van Quyen 2016; Maynard et al., 1997; Woepfel et al., 2021). These five I stumbled upon by chance, better please doublecheck all citations in the text.

Reviewers' comments:

Reviewer #3 (Remarks to the Author):

This revised manuscript is greatly improved. The authors have addressed the concerns, and the revisions clarify many issues.

One concern relates to the degree that extracellular negativity always corresponds to net synaptic excitation, and vice versa with the positivity always corresponding to synaptic inhibition. The authors have attempted to address the concern with reference to their earlier work.

There is still a concern about whether the data analyses and conclusions will be fully comprehensible to a wide audience, but the authors have made a significant and good-faith attempt to address these concerns.

Reviewer #4 (Remarks to the Author):

This is a revised and improved manuscript. The authors have provided detailed and extensive responses to original critique with corresponding changes to the text and figures. Questions were raised in original review regarding the interpretation of field potential polarity recorded at the cortical surface in regards to underlying excitation and inhibition. These questions remain. Negative field potentials are assumed to reflect net excitation and positive potentials, inhibition. This is certainly a possible and plausible interpretation but is by no means indicated by the data. The authors refer to a paper by Eissa et al. 2017, as justification for the interpretation. This paper includes good discussion of dipole modeling as it relates to surface potentials, and more importantly, relates increased MUA in these specific data to field potential negativity. Thus, as noted by the authors, there is support for their interpretations of excitation and inhibition in the specific context of these data. Yet, even with extensive and thoughtful edits for clarity, the paper remains highly specialized. While this could limit the audience, the results should be interesting and important to those with requisite expertise.

Reviewer #5 (Remarks to the Author):

As I am not the one who raised the original points it is a bit hard for me to judge the response to other people's comments (I assume the original reviewer #1 was not available anymore?) but from my point of view the authors have responded to all these points in a very detailed and comprehensive manner. The same holds for the responses to the other three reviewers but again I do not think I should be the one to judge this.

But as an independent voice I would say that the revised paper (I haven't seen the original paper) has become quite clear and understandable and from my point of view deserves to be published in Communications Biology.

However, as I also did my own reading of this revised version I do have a couple of (mostly minor) additional comments:

Very first sentence: maybe "can reflect" (not only, they also reflect non-pathological activity). Or rephrase somehow.

The text in the Abstract (lines 39-40) was modified to the following:

"The electrographic manifestation of neural activity can reflect the relationship between the faster action potentials of individual neurons and the slower fluctuations of the local field potential (LFP)."

Fig. 2: It is quite obvious that after translation to the origin the grid locations close to the center have a much higher support than those at the edges (it might be useful to get to see a support matrix for a typical example). This will lead to different error bars for the average and maybe even different expectation values (following a radial dependence) and the question comes to mind whether this can lead to statistical irregularities, in particular since later data are transformed to polar coordinates where the r-dependence is investigated specifically. Would it be possible to use simulations to somehow check the validity of the results (how much of the radial symmetry could be due to this effect), e.g. by assuming a constant number of spikes per pixel (with uniform spatiotemporal data and some controlled randomization) and observing the resulting r-dependence. A similar kind of "surrogate" analysis is performed in Fig. 6B.

Indeed, spikes located closer to the center of the MEA will have a higher contribution to the averaged st-SCA in comparison to spikes located to the periphery of the MEA as there are more spikes near the center. For this reason, we have employed several validation metrics (please see Figures S6 & S7). We first randomized the spike timing across the MEA, which yielded in loss of the spatiotemporal patterns observed. We then decorrelated the LFP signals by applying a spatial filter across the channels, which did not significantly change the spatiotemporal patterns. We have now updated Figures S6 & S7 to include the average signal-to-noise ratios (SNR). The captions of these figures have been updated to include the following text (lines 180-181 & lines 185-186):

"The average signal-to-noise ratios (SNR) are listed per patient."

In addition, we added two tables to the Supplemental Materials (Table S3, lines 194-195; Table S4, lines 196-197): Table S3 includes a typical example of the 19x19 grid weights, and Table S4 includes the corresponding SNR values per grid position obtained from the st-SCA calculation across the MEA. With few exceptions, the SNR at each position of the 19x19 grid satisfies the Rose criterion stating that a reliable signal amplitude of an image

is at least 4-5 times the noise amplitude (SNR 12-14dB) (Rose, 1973; Bushberg et al., 2012). The values below the Rose criterion in our images were the reason that we do not analyze or depict the corners of the images of the st-SCA. We have updated the Methods section of the Supplemental Materials to clarify this point (lines 105-113). The text is repeated below for convenience:

“A typical example of the total number of contributions per each position in the 19x19 grid is shown in Table S3. The average values in each pixel of the 19x19 image is based on variable numbers of trials, with less trials towards the corners of the picture (e.g., Table S3). The signal-to-noise ratio of these pixels is listed in Table S4. With the exception of very few pixels (corners in Table S4), all locations satisfy the so-called Rose criterion, namely that to distinguish image features reliably, the signal’s amplitude must be 4-5 times (12-14 dB) the amplitude of the associated noise (Rose, 1973; Bushberg et al., 2012). In the images we excluded the (corner) pixel values that do not satisfy the Rose criterion (including the NaN – Not a Number – in Table S4).”

Figs. 3, 4, and 6: Why is always patient 1 used (for example the first plot in both of these Figures is actually identical)? If results hold in a more general manner it should be possible to include more variation there (or if you would like to be consistent there show the results for the other patients (the ones relevant here) in the SM. The way it is now it somehow gives the impression that P1 is the model patient and for the others things do not work out as well (and I mean even the others that show temporal and spatial patterns consistent with a symmetric relationship).

We show the results for the representative st-SCAs for each Patient in Figure S2. Patients 1-3 resemble sinc functions, and Patients 4-5 resemble deep wells of excitation. We updated the following text to the Results section of the main text (line 255) to clarify this:

“Representative st-SCAs for each patient are depicted in Fig. S2.”

References:

Many references seem to be missing, e.g. all the ones cited at the end of the Discussion (e.g Cotterill et al., 2016; Kapucu et al., 2022; Le van Quyen 2016; Maynard et al., 1997; Woepel et al., 2021). These five I stumbled upon by chance, better please double check all citations in the text.

We thank the reviewer for pointing out this oversight. The missing references have been added, and the reference list has been double checked for completeness.

REVIEWERS' COMMENTS:

Reviewer #5 (Remarks to the Author):

The authors have addressed all of my concerns and I can now recommend the article for publication in Communications Biology.